# Kinetic regulation of kinesin's two motor domains coordinates its stepping along microtubules

**Yamato Niitani[1†], Kohei Matsuzaki[1,2†], Erik Jonsson[3], Ronald D Vale[3], Michio Tomishige[1,2*]**

[1]Department of Applied Physics, School of Engineering, The University of Tokyo, Tokyo, Japan; [2]Department of Physical Sciences, College of Science and Engineering, Aoyama Gakuin University, Sagamihara, Japan; [3]Howard Hughes Medical Institute and Department of Cellular and Molecular Pharmacology, University of California, San Francisco, San Francisco, United States

## eLife Assessment

This study provides **compelling** evidence that kinesin's stepping mechanism is governed by strain-induced conformational changes in its nucleotide-binding pockets. Using pre-steady state kinetics and single-molecule assays, the authors demonstrate that the neck linker's conformation differentially modulates nucleotide affinity and detachment rates, establishing an asynchronous chemo-mechanical cycle that prevents simultaneous detachment. Supported by cryo-EM structural data, the work presents an **important** advance in our understanding of kinesin's hand-over-hand movement. [Editors' note: this paper was reviewed by Review Commons.]

*For correspondence:
tomishige@phys.aoyama.ac.jp

[†]These authors contributed equally to this work

**Abstract** The two identical motor domains (heads) of dimeric kinesin-1 move in a hand-over-hand process along a microtubule, coordinating their ATPase cycles such that each ATP hydrolysis is tightly coupled to a step and enabling the motor to take many steps without dissociating. The neck linker, a structural element that connects the two heads, has been shown to be essential for head–head coordination; however, which kinetic step(s) in the chemomechanical cycle is 'gated' by the neck linker remains unresolved. Here, we employed pre-steady-state kinetics and single-molecule assays to investigate how the neck-linker conformation affects kinesin's motility cycle. We show that the backward-pointing configuration of the neck linker in the front kinesin head confers higher affinity for microtubule, but does not change ATP binding and dissociation rates. In contrast, the forward-pointing configuration of the neck linker in the rear kinesin head decreases the ATP dissociation rate but has little effect on microtubule dissociation. In combination, these conformation-specific effects of the neck linker favor ATP hydrolysis and dissociation of the rear head prior to microtubule detachment of the front head, thereby providing a kinetic explanation for the coordinated walking mechanism of dimeric kinesin.

## Introduction

Kinesin-1 (hereafter called kinesin) is a dimeric motor protein that transports intracellular cargo by moving along microtubules. The unidirectional motion of kinesin toward the plus-end of the microtubule is driven by an N-terminal globular domain (termed 'head') that catalyzes ATP hydrolysis and binds to the microtubule. The head is followed by a 14 amino acid segment called the 'neck linker' and then a long coiled-coil stalk that dimerizes the two identical polypeptide chains. Dimeric kinesin

takes an 8-nm center-of-mass step by moving the rear head past the forward head along a micro-tubule protofilament once per ATP hydrolysis cycle; by alternating the movement of its two heads, kinesin can move processively for a micron or more without dissociating (*Vale and Milligan, 2000*; *Asenjo et al., 2003*; *Kaseda et al., 2003*; *Asbury et al., 2003*; *Yildiz et al., 2004*; *Isojima et al., 2016*; *Wolff et al., 2023*). The cycle of binding and dissociation of the head to/from the microtubule is tightly correlated with the ATPase cycle; release of ADP, creating a nucleotide-free state, stabilizes the microtubule binding of the head, while phosphate release after ATP hydrolysis induces detachment of the head from the microtubule (*Gilbert and Johnson, 1994*; *Ma and Taylor, 1997a*; *Cross, 2004*). Therefore, to alternately move the two heads, the ATP hydrolysis cycles in the rear and/or front heads should be differentially regulated (gated), such that the rear head hydrolyzes ATP and detaches from the microtubule, prior to the front head (*Hackney, 1994*; *Ma and Taylor, 1997b*; *Gilbert et al., 1998*; *Auerbach and Johnson, 2005a*; *Cochran, 2015*).

To gate the ATPase cycles during processive movement, there must be a structural mechanism that enables the two heads to communicate with one another. When both heads are bound to the adjacent tubulin-binding sites (two-head-bound state), the front and rear heads are separated by 8 nm and cannot directly interact with each other. However, their neck linkers are highly extended and adopt two distinct configurations: the neck linker in the front head is stretched backward, and the neck linker in the rear head is pointing forward, favoring the docked state (*Figure 1A*; *Rice et al., 1999*; *Skiniotis et al., 2003*; *Tomishige et al., 2006*; *Liu et al., 2017*). Artificially extending the neck linkers by inserting polypeptide spacers reduces the coupling ratio between the forward step and ATP turnover, indicating that the strain transmitted through the neck linkers has an essential role in head–head coordination (*Hackney et al., 2003*; *Yildiz et al., 2008*; *Shastry and Hancock, 2010*; *Andreasson et al., 2015*; *Isojima et al., 2016*).

Several models, which are not mutually exclusive, have been put forth to explain the alternating catalysis of the two heads and the high coupling between ATP turnover and stepping (*Block, 2007*; *Valentine and Gilbert, 2007*; *Gennerich and Vale, 2009*). One type of model proposes neck linker-based regulation of microtubule affinity. A 'rear-head gating' version of this model proposes that forward strain posed through the neck linker accelerates the detachment of the rear head. Evidence supporting this model is based on experiments showing that the two-headed kinesin hydrolyzes ATP and detaches from microtubule much faster than a one-headed kinesin (*Hancock and Howard, 1999*; *Crevel et al., 2004*; *Schief et al., 2004*) and that the detachment rate of the head from microtubule is faster under a forward than a rearward pulling force from an optical trap (*Uemura et al., 2002*). However, other experimental results showed that the detachment rate from the microtubule by the rear head of a dimer or of the monomer under assisting load was not significantly accelerated (*Rosenfeld et al., 2003*; *Andreasson et al., 2015*; *Dogan et al., 2015*).

Another class of gating models proposes that nucleotide state transition in the ATPase cycle is sensitive to neck-linker tension between the two heads. A 'front-head gating' model has been proposed that tension in the two-head-bound state decreases ATP binding in the front head (*Rosenfeld et al., 2003*; *Klumpp et al., 2004*; *Guydosh and Block, 2006*; *Dogan et al., 2015*). Pre-steady-state kinetic measurements showed that binding of mant-ATP (a fluorescent analog that shows a signal change upon kinesin binding) to a microtubule-docked kinesin dimer showed single exponential kinetics with a signal amplitude nearly half than that expected if both heads can bind ATP (*Rosenfeld et al., 2003*; *Klumpp et al., 2004*). However, interpreting these results is complex, since the amplitude and duration of the mant-ATP signal is a composite from the front and rear heads, which complicates the assignment of specific rate constants of nucleotide binding and unbinding or isomerization events to the two microtubule-bound heads.

In this study, we measured the ATP-binding and microtubule-detachment kinetics (*Figure 1B*) of the front and rear heads separately using pre-steady-state and single-molecule measurements. To isolate and measure the kinetics of the front and rear head states separately, we utilized two methods: (1) disulfide-crosslinking to immobilize the neck linker in monomeric kinesin in the forward- or backward-pointing configurations (mimicking the rear and front head states, respectively), and (2) a heterodimeric kinesin in which one polypeptide chain was locked in neck linker docked rear head conformation. Our results provide quantitative evidence that both kinesin heads participate in a coordinated gating mechanism: the backward constraint of the neck linker in the front head reduces its detachment rate from microtubules without changing ATP on/off rates, while the forward constraint

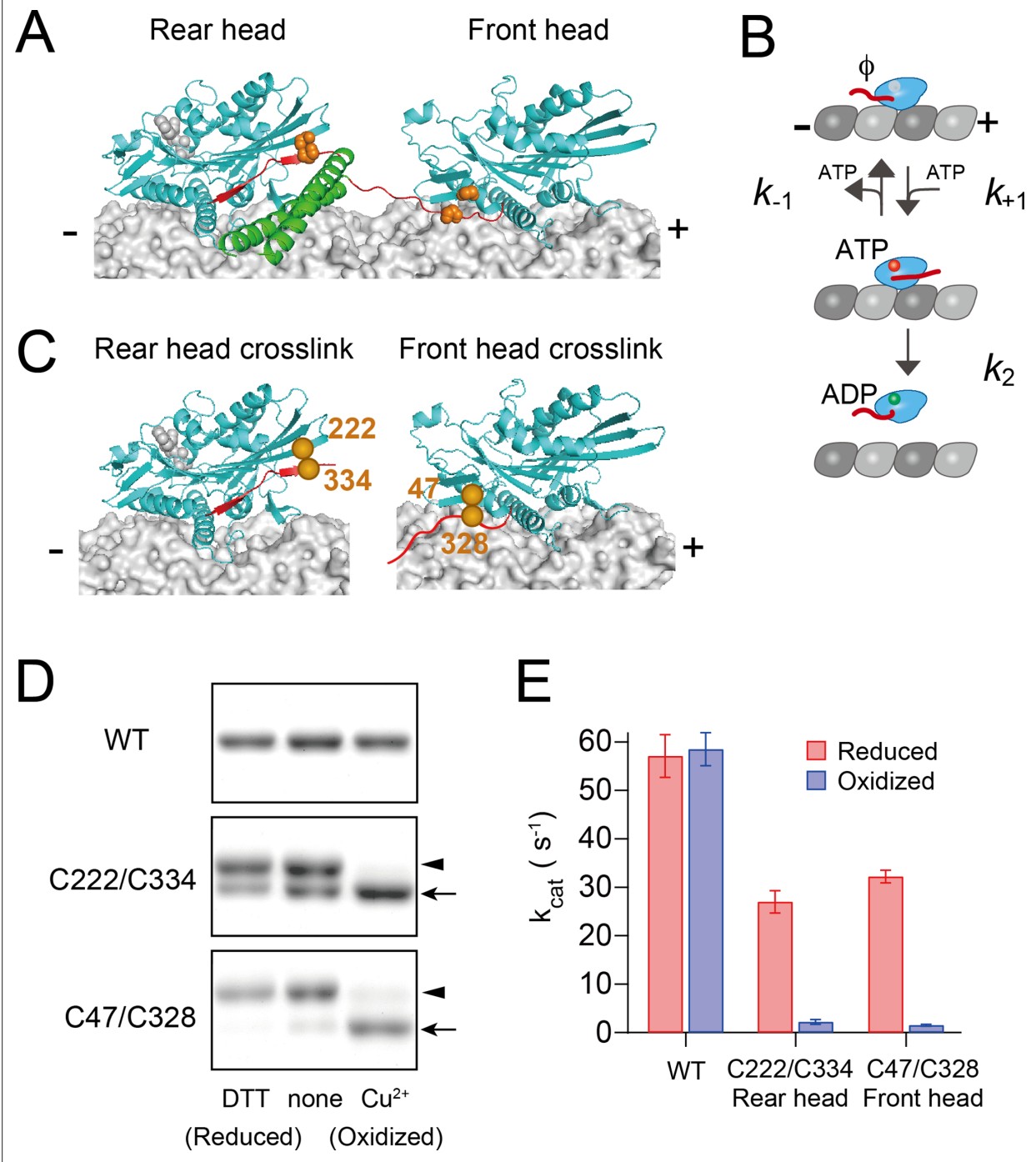

**Figure 1.** Disulfide-crosslinking to constraint the neck linker in either the forward- or backward-pointing configuration. (**A**) Three-dimensional structure of kinesin dimer on the microtubule in the two-head-bound intermediate state (modeled based on PDB# 4HNA and 4LNU). The motor domain, neck linker, and neck coiled-coil are shown in cyan, red, and green. Orange spheres represent the positions of cysteine residues as described in panel (**C**). (**B**) Michaelis–Menten kinetics scheme for ATP-promoted dissociation of kinesin head from the microtubule, which includes ATP-binding ($k_{+1}$), ATP dissociation ($k_{-1}$), and ATP-induced microtubule dissociation ($k_2$) steps. (**C**) Positions of the cysteine residues for disulfide-crosslinking of monomeric kinesin. Cys222 and Cys334 were introduced to constrain the neck linker in the forward, docked conformation (left), while Cys47 and Cys328 were introduced to constrain the neck linker in the backward-pointing configuration (right). (**D**) Disulfide-crosslinking after oxidative treatment was analyzed by non-reducing SDS–PAGE. WT (without Cys introduction), C222/C334, and C47/C328 constructs were treated with a reducing reagent (DTT), none, or oxidizing reagents ($Cu^{2+}$). Arrowheads and arrows indicate un- and intramolecular-crosslinked kinesins, respectively. The whole image of the gels is shown in *Figure 1—figure supplement 1*. (**E**) Microtubule-activated ATPase under reduced and oxidized conditions. $k_{cat}$ and $K_m$(MT) were determined

*Figure 1 continued on next page*

*Figure 1 continued*

from the fit to a Michaelis–Menten equation of ATP turnover rates at various microtubule concentrations (*Figure 1—figure supplement 2*). Mean ± SEM values were obtained from three independent assays.

The online version of this article includes the following source data and figure supplement(s) for figure 1:

**Source data 1.** JPG files containing original gels for *Figure 1D*, indicating the relevant bands and treatments.

**Source data 2.** Original files for SDS–PAGE displayed in *Figure 1D*.

**Figure supplement 1.** Estimation of disulfide-crosslinking efficiency.

**Figure supplement 1—source data 1.** JPG files containing original gels for *Figure 1—figure supplement 1A*, indicating the relevant bands and treatments.

**Figure supplement 1—source data 2.** Original files for SDS–PAGE displayed in *Figure 1—figure supplement 1A*.

**Figure supplement 2.** Steady-state microtubule-activated ATPase rates of monomeric kinesins before and after oxidative crosslinking.

**Figure supplement 2—source data 1.** Excel file containing ATPase measurement data for *Figure 1—figure supplement 2* and *Figure 3—figure supplement 4A*.

on the neck linker in the rear head dramatically reduces ATP dissociation rate but has minimal effect on microtubule affinity. These results are consistent with the open and closed conformations of the nucleotide-binding pocket in the front and rear heads of microtubule-bound kinesin dimers observed in cryo-electron microscopy (cryo-EM) studies (*Liu et al., 2017*; *Benoit et al., 2021*) and provide a mechanism for the coordinated hand-over-hand movement of kinesin.

## Results
### Controlling the neck-linker orientation of monomer kinesin through crosslinking

We first sought to examine the kinetics of nucleotide binding and microtubule dissociation of monomeric kinesin in which the neck linker was oriented either in the forward (mimicking the rear head) or the rearward (mimicking the front head) direction (*Figure 1A, C*). To control the orientation of the neck linker in monomer kinesin, we employed disulfide-crosslinks between two cysteines, one on the head and the other on the neck linker (*Tomishige and Vale, 2000*). To immobilize the neck linker in this forward-pointing, docked conformation (*Gigant et al., 2013*), K222 (located at the plus-end oriented tip of the head) and E334 (the distal end of the docked neck linker) residues were substituted with cysteines in a Cys-light monomeric kinesin (K339 Cys-light), whose solvent-exposed cysteines were replaced by either Ser or Ala (*Rice et al., 1999*; *Figure 1C*, left). We refer to the C222/C334 crosslink here as 'rear head crosslink'. We previously introduced this cysteine pair into dimeric kinesin and found that oxidative crosslinking resulted in a complete loss of directionality (*Tomishige and Vale, 2000*). The neck linker of the front head is detached from the head, and its distal end is pulled backward (*Skiniotis et al., 2003*; *Tomishige et al., 2006*; *Benoit et al., 2021*). To constrain the neck linker in this backward-pointing configuration, which we refer to as the 'front head crosslink', A47 (the minus-end-oriented base of the head) and T328 (five residues from the proximal end of the neck linker) residues were substituted with cysteines (*Figure 1C*, right). The cysteine pairs were covalently crosslinked by oxidative treatment with copper ion and *o*-phenanthroline (*Kobashi, 1968*; *Kaan et al., 2011*). The crosslinking efficiencies were estimated, by running SDS–PAGE under non-reducing conditions, to be 97% and 87% for the rear and front head crosslinks, respectively (*Figure 1D*, *Figure 1—figure supplement 1*).

We first evaluated the effect of disulfide-crosslinking on the microtubule-activated ATP turnover rates (*Figure 1—figure supplement 2*). Under reducing conditions, the $k_{cat}$ for both the rear and front head crosslinks (27.0 and 32.2 ATP/s per head, respectively) were lower than WT (57.1 ATP/s per head) (*Figure 1E*). The reduced ATPase activity primarily results from a decreased microtubule association rate (data to be presented elsewhere) with little change in ATP binding or microtubule dissociation rates (*Table 1*). For the rear head crosslink, the reduced ATPase activity might also be due to the presence of crosslinked species even under the reducing condition (~30%; *Figure 1D*, *Figure 1—figure supplement 1B*). Under oxidized conditions, the $k_{cat}$ values for both rear and front head crosslinks were reduced by more than 10-fold compared to that under reducing conditions (2.2 and 1.5 ATP/s

**Table 1.** Kinetic rate constants for disulfide-crosslinking monomers.

Steady-state ATP-turnover rate ($k_{cat}$ and $K_m$(MT)), pre-steady-state mant-ATP binding ($k_{+1}$ and $k_{-1}$), and microtubule dissociation rates ($k_2$ and $k_{2\_SMF}$, measured by light scattering and single-molecule fluorescence) measured under reduced and oxidized conditions.

| | WT | | C222/C334 | | C47/C328 | |
|---|---|---|---|---|---|---|
| | Reduced | Oxidized | Reduced | Oxidized | Reduced | Oxidized |
| $k_{cat}$ (s$^{-1}$) | 57.1 ± 4.4 | 58.5 ± 3.4 | 27.0 ± 2.3 | 2.2 ± 0.5 | 32.2 ± 1.3 | 1.5 ± 0.2 |
| $K_m$(MT) (μM) | 0.31 ± 0.08 | 0.36 ± 0.08 | 0.23 ± 0.06 | 0.200 ± 0.007 | 0.28 ± 0.06 | 0.13 ± 0.04 |
| $k_{+1}$ (s$^{-1}$ μM$^{-1}$) | 5.5 ± 0.1 | 6.0 ± 0.4 | 5.1 ± 0.7 | 1.62 ± 0.07 | 4.0 ± 0.2 | 2.0 ± 0.1 |
| $k_{-1}$ (s$^{-1}$) | 136 ± 5 | 125 ± 12 | 117 ± 23 | 3.7 ± 1.9 | 96 ± 6 | 42 ± 4 |
| $k_2$ (s$^{-1}$) | 52.1 ± 0.6 | 59.1 ± 0.9 | 69.3 ± 0.5 | N.D. | 51.5 ± 0.7 | 7.0 ± 0.3 |
| $k_{2\_SMF}$ (s$^{-1}$) | 44.0 ± 1.9 | 41.6 ± 1.2 | 47.2 ± 1.2 | 57.1 ± 1.8 | 42.1 ± 0.4 | 2.30 ± 0.08 |

| | | | C4/C330 | | C47/C335 | |
|---|---|---|---|---|---|---|
| | | | Reduced | Oxidized | Reduced | Oxidized |
| | | $k_{cat}$ (s$^{-1}$) | 33.5 ± 2.1 | 32.6 ± 1.2 | 43.1 ± 3.0 | 11.9 ± 0.6 |
| | | $K_m$(MT) (μM) | 0.42 ± 0.08 | 8.36 ± 0.98 | 0.33 ± 0.05 | 0.10 ± 0.01 |
| | | $k_{+1}$ (s$^{-1}$ μM$^{-1}$) | 4.7 ± 0.5 | 4.1 ± 0.3 | 9.1 ± 0.3 | 4.5 ± 0.1 |
| | | $k_{-1}$ (s$^{-1}$) | 148 ± 18 | 96 ± 9 | 85 ± 10 | 41 ± 5 |
| | | $k_2$ (s$^{-1}$) | 68.6 ± 0.5 | 80.8 ± 0.8 | 41.6 ± 0.6 | 19.4 ± 0.2 |
| | | $k_{2\_SMF}$ (s$^{-1}$) | 59.5 ± 2.3 | 58.4 ± 2.6 | 52.1 ± 2.0 | 22.5 ± 0.3 |

per head, respectively (*Figure 1E* and *Table 1*)). These results demonstrate that neck linker constraints in both forward and rearward orientations inhibit specific steps in the mechanochemical cycle of the head.

## Effect of neck linker crosslinking on the binding/dissociation rates of ATP

To measure ATP-binding rate to the kinesin head on the microtubule, we rapidly mixed the nucleotide-free kinesin–microtubule complex with increasing concentrations of mant-ATP using stopped-flow apparatus. The initial rapid increase in the fluorescence represents mant-ATP binding to the nucleotide pocket of the motor domain; this rapidly increasing phase was fit with an exponential to obtain the observed rate $k_{obs}$ (*Figure 2—figure supplement 1*). At low mant-ATP concentrations, the $k_{obs}$ increased linearly as a function of mant-ATP concentration (*Ma and Taylor, 1997a*; *Moyer et al., 1998*); the slope of the fitted line provides the binding rate $k_{+1}$ and the offset provides the dissociation rate $k_{-1}$ (*Figure 2A*). First, we demonstrated that $k_{+1}$ and $k_{-1}$ of the wild-type head without Cys-modification were unchanged after oxidization (*Table 1*) and were comparable to those previously reported (*Cross, 2004*). Rear head crosslinking resulted in a modest decrease in the on-rate $k_{+1}$ by threefold (5.1 and 1.6 s$^{-1}$ μM$^{-1}$ under reducing and oxidizing conditions, respectively; *Figure 2B*). However, rear head crosslinking produced a 30-fold decrease in the off-rate $k_{-1}$ (117 and 3.7 s$^{-1}$ under reducing and oxidizing conditions, respectively). In contrast, front head crosslinking resulted in only a modest and similar twofold decrease in both $k_{+1}$ and $k_{-1}$ ($k_{+1}$ was 4.0 and 2.0 s$^{-1}$ μM$^{-1}$, $k_{-1}$ was 96 and 42 s$^{-1}$ under reducing and oxidizing conditions, respectively; *Figure 2B*), resulting in little or no change in the affinity for ATP after crosslinking. These results indicate that the rear head crosslink reduces the ATP dissociation rate, while the front head crosslink does not significantly alter the ATP-binding kinetics. We note that the measured $k_{-1}$ of the rear head crosslink was similar to the ATP turnover rate $k_{cat}$ (*Table 1*); therefore, it is likely that the majority of ATP molecules bound to the head were hydrolyzed, and the observed fluorescence decrease rather represents mant-ADP release.

To detect the ATP dissociation event of the rear head, we employed a mutant kinesin with a point mutation of E236A in the switch II loop, which almost abolishes ATPase hydrolysis and traps in the microtubule-bound, neck-linker docked state (*Rice et al., 1999*; *Auerbach and Johnson, 2005b*). The

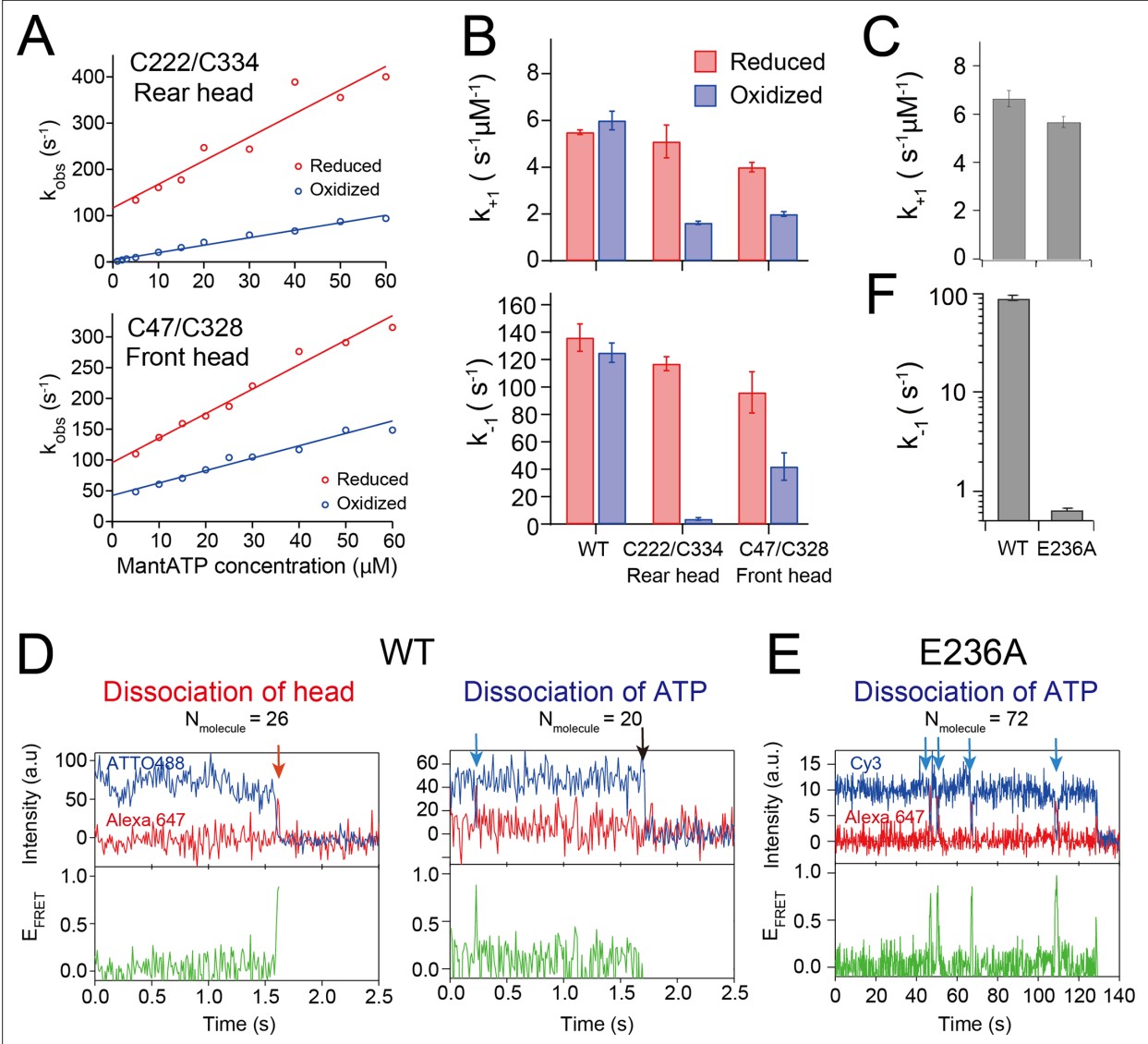

**Figure 2.** ATP-binding/dissociation kinetics of disulfide-crosslinked monomers. (**A**) The rate of initial enhancement of the fluorescence after rapid mixing of kinesin–microtubule complex with mant-ATP ($k_{obs}$; *Figure 2—figure supplement 1*) was plotted as a function of mant-ATP concentration. Solid lines represent a linear fit; the slope provides the on-rate of mant-ATP ($k_{+1}$) and the off-set provides the off-rate of mant-ATP ($k_{-1}$). (**B**) ATP-binding rate $k_{+1}$ and ATP dissociation rate $k_{-1}$ of monomers before and after crosslinking obtained from the fit shown in pane (**A**). Error bar shows the fit error. (**C**) Mant-dATP-binding rate $k_{+1}$ to wild-type (WT) and E236A monomers ($k_{obs}$ plots are shown in *Figure 2—figure supplement 2*). (**D**) Two typical traces of single-molecule fluorescence resonance energy transfer (smFRET) between the donor-labeled wild-type monomer and Alexa-ATP (500 nM). Fluorescence intensities of donor (blue) and acceptor (red), and the calculated FRET efficiency (green) recorded at 100 fps are shown. Left trace: the FRET efficiency transiently increases just before both dyes disappear (indicated by the red arrow), suggesting that the nucleotide-bound head detached from the microtubule (*Figure 2—figure supplement 4A*). Right trace: the FRET efficiency transiently increases, followed by the recovery of donor fluorescent (shown by the blue arrow), indicating the reversible dissociation of ATP. The black arrow indicates the photobleaching of the donor dye. The numbers of molecules observed for each type is shown at the top of the traces. (**E**) Typical trace of smFRET between the E236A monomer and Alexa-ATP (200 nM) recorded at 5 fps, demonstrating repetitive ATP binding and dissociation. (**F**) The dissociation rate $k_{-1}$ of Alexa-ATP from wild-type and E236A monomers (histograms are shown in *Figure 2—figure supplement 4B*). The value for the wild-type may be underestimated due to the limited temporal resolution (10 ms).

The online version of this article includes the following source data and figure supplement(s) for figure 2:

**Source data 1.** Excel file containing $k_{obs}$ plot data of mant-ATP binding for *Figure 2A* and *Figure 3—figure supplement 4B*.

**Source data 2.** Excel file containing fluorescent intensity time traces of donor and acceptor for *Figure 2D, E*.

**Figure supplement 1.** Pre-steady-state kinetics of ATP binding to monomeric kinesin–microtubule complex before and after crosslinking.

**Figure supplement 1—source data 1.** Excel file containing mant-ATP-binding trace data for *Figure 2—figure supplement 1*.

*Figure 2 continued on next page*

*Figure 2 continued*

**Figure supplement 2.** ATP-binding kinetics for E236A mutant monomer on microtubule.

**Figure supplement 2—source data 1.** Excel file containing mant-ATP-binding trace data for *Figure 2—figure supplement 2A*.

**Figure supplement 3.** Processive motility of wild-type kinesin dimer driven by Alexa 647 ATP.

**Figure supplement 3—source data 1.** Excel file containing single-molecule fluorescent motility data for *Figure 2—figure supplement 3B*.

**Figure supplement 4.** Dwell time of Alexa-ATP-bound state for wild-type and E236A monomers determined using single-molecule fluorescence resonance energy transfer (smFRET).

**Figure supplement 4—source data 1.** Excel file containing single-molecule FRET histogram data for *Figure 2—figure supplement 4B*, *Figure 4—figure supplement 3*, and *Figure 6—figure supplement 5B*.

mant-ATP-binding kinetics of E236A monomer revealed an ATP on-rate ($k_{+1}$ of 5.6 s$^{-1}$ μM$^{-1}$) comparable to that of wild-type monomer (6.6 s$^{-1}$ μM$^{-1}$), indicating that the initial ATP-binding step remains unaffected by the mutation (*Figure 2C*, *Figure 2—figure supplement 2*). The ATP off-rate could not be precisely determined from the offset value of the linear fitting of the $k_{obs}$ plot due to its small magnitude.

To determine the ATP off-rate $k_{-1}$ of E236A monomer, we employed single-molecule fluorescence resonance energy transfer (smFRET) between the donor dye attached to the head and the acceptor dye conjugated to ATP. The residue S55, separated by ~0.8 nm from the nucleotide pocket, was substituted with cysteine and was labeled with ATTO488 or Cy3 dye; we would expect ~90% high FRET efficiency when Alexa 647-conjugated ATP (termed as Alexa-ATP) binds to the labeled head (*Figure 2—figure supplement 4A*). Using a wild-type kinesin dimer, we demonstrated that Alexa-ATP acts as a substrate to drive kinesin motility along microtubules, although velocities were about 10-fold lower than those driven by unmodified ATP (*Figure 2—figure supplement 3*). First, we observed single-molecule FRET of ATTO488-labeled 'wild-type' monomer on microtubule in the presence of 500 nM Alexa-ATP. We observed FRET efficiency increases upon Alexa-ATP binding to microtubule-bound monomer, and then the donor and acceptor simultaneously disappeared (representing that the head dissociated from microtubule after ATP hydrolysis; *Figure 2D*, left), or the donor fluorescence recovered after the disappearance of the acceptor fluorescence (representing that ATP dissociated while the head remains bound to the microtubule; *Figure 2D*, right). We found that the ATP bound to the wild-type head could be either hydrolyzed, followed by head dissociation from the microtubule, or reversibly dissociated with nearly equal probability (*Figure 2D*). This finding is consistent with the mant-ATP-binding kinetics measurements of the monomer without neck-linker constraint, which showed similar $k_2$ and $k_{-1}$ values (*Table 1*). In contrast, the single-molecule FRET observation of Cy3-labeled E236A monomer in the presence of 200 nM Alexa-ATP showed that the majority of the Cy3-labeled mutant remained bound to the microtubule during the observation period, and Alexa-ATP molecules repeatedly bound and reversibly dissociated to/from the mutant head (*Figure 2E*). The mean dwell time of the high FRET state (ATP-bound state) was 1.54 s, which corresponds to $k_{-1}$ of 0.65 s$^{-1}$ (*Figure 2F*, *Figure 2—figure supplement 4B*), a value 140-fold smaller than that of the wild-type monomer (91 s$^{-1}$). We note that the neck linker of the E236A monomer was neither pulled forward nor constrained by crosslinking; therefore, the ATP off-rate of monomeric E236A may still be faster than that of the rear head in dimeric kinesin (as will be examined later).

## Effect of neck linker crosslinking on the dissociation rate from microtubule

Next, we analyzed the effect of the neck-linker crosslinking on the detachment rate of the monomeric kinesin from the microtubule (MT dissociation). First, we measured turbidity change after rapidly mixing the nucleotide-free kinesin-microtubule complex with 1 mM ATP using a stopped-flow apparatus. Light scattering signals, which vary with particle size, have been established as a method for determining microtubule dissociation kinetics (*Ma and Taylor, 1997a*). Since monomeric kinesin repeatedly hydrolyzes ATP per encounter to the microtubule through electrostatic interactions (*Jiang and Hackney, 1997*), we included 150 mM KCl to prevent reassociation of the detached head. However, the rear head crosslink kinesin did not show any initial burst in turbidity after oxidative treatment (*Figure 3—figure supplement 1*) because this constraint on the neck linker dramatically reduces the microtubule-activated ADP release rate (data to be presented elsewhere), creating a

weak microtubule binding state. In contrast, the front head crosslink kinesin displayed an initial burst phase for MT dissociation under these salt conditions. The fit of the initial decay revealed a sevenfold decrease in the MT dissociation rate $k_2$ (51.5 and 7.0 s$^{-1}$ for reduced and oxidized conditions, respectively (*Figure 3A*)).

Since we could not measure the MT dissociation for the rear head crosslink using light scattering, we sought an alternate approach involving visualizing GFP-fused K339 Cys-light kinesin binding to the microtubule using single-molecule fluorescent microscopy in the presence of 100 mM KCl and 1 mM ATP (*Figure 3B*). The rear head crosslink showed a sixfold reduced binding frequency to the microtubule (*Figure 3—figure supplement 2A*). We fit the histograms of the dwell time of fluorescent spots on microtubules with single exponential decay (*Figure 3—figure supplement 2B*) and employed the inverse of the dwell time as a rough estimate of the dissociation rate (termed as $k_{2\_SMF}$). The $k_{2\_SMF}$ of the rear head crosslink was slightly increased compared to before crosslinking (47.2 and 57.1 s$^{-1}$ under reducing and oxidizing conditions, respectively). In contrast, the front head crosslink showed an 18-fold decrease in the $k_{2\_SMF}$ compared to before crosslinking (42.1 and 2.3 s$^{-1}$ under reducing and oxidizing conditions, respectively) (*Figure 3C* and *Table 1*). These results suggest that forward constraint on the neck linker in the rear head does not significantly accelerate the detachment from the microtubule, while backward constraint of the neck linker in the front head reduces the detachment rate.

## Different crosslinking schemes to examine the role of the neck-linker docking on gating

The reduced detachment rate of the front head crosslink is likely due to the inability of the C47/C328 crosslinked head to adopt the neck-linker docked conformation. Then, we asked whether the reduced detachment rate can be recovered if the crosslinked neck linker is allowed to partially dock onto the head. To test this, we crosslinked the neck linker in the backward orientation between C47 on the head and C335 on the neck linker (*Figure 3D*, right), which produces an extra seven amino acids compared to C47/C328 between the base and the crosslinked residues on the neck linker, allowing the initial few residues of the neck linker (including I325 residue involved in filling the hydrophobic pocket exposed on the ATP-bound head; *Vale and Milligan, 2000*; *Sindelar, 2011*; *Cao et al., 2014*) to dock onto the head (*Figure 3—figure supplement 3*). The ATP-induced detachment rate $k_2$ of C47/C335 showed only a twofold reduction after crosslinking (41.6 and 19.4 s$^{-1}$ for reduced and oxidized conditions, respectively, measured using light scattering; *Figure 3A, C*, *Figure 3—figure supplement 4C–E*). This finding is consistent with the idea that the gatekeeper of the front-head gate is the neck-linker docking and suggests that partial neck-linker docking onto the head is sufficient to half open the gate, while more extensive neck-linker docking is required to fully open the gate.

In contrast, the neck-linker orientation in the rear head crosslink did not change the dissociation rate from microtubules but reduced the dissociation rate of ATP (*Figure 2B*). We examined an alternate crosslinking scheme between C4 on the N-terminal stretch and C330 on the neck linker (*Tomishige and Vale, 2000*), which couples the neck linker with the N-terminal flexible segment called 'cover strand (CS)' (*Figure 3D*, left). The interaction between the neck linker and CS has been shown to be important to produce a forward bias in the diffusional motion of the detached head against hindering load (*Hwang et al., 2008*; *Khalil et al., 2008*). The ATP dissociation rate $k_{-1}$ was not significantly affected by C4/C330 crosslinking (148 and 96 s$^{-1}$ for reduced and oxidized conditions, respectively; *Figure 3E* and *Figure 3—figure supplement 4B*), indicating that the interaction of the neck linker with the head, not with the N-terminal CS, is responsible for the reduction of the ATP off-rate.

## An E236A–WT heterodimer to distinguish front and rear heads

The front head crosslink was designed to constrain the neck linker to mimic the neck-linker orientation of the front head. However, it differs from that of the front head in that the backward-pointing configuration is maintained by the crosslinking, not by the backward strain originating from the stretched neck linkers between two microtubule-bound heads (*Hyeon and Onuchic, 2007*; *Hariharan and Hancock, 2009*). We sought to further investigate how the neck-linker tension contributes to the front-head gating by examining the nucleotide-binding and microtubule-detachment kinetics of the front head in the context of a dimeric kinesin. To this end, we employed a heterodimer in which one polypeptide is wild-type (WT) and the other contains the E236A mutation (E236A–WT heterodimer). As described

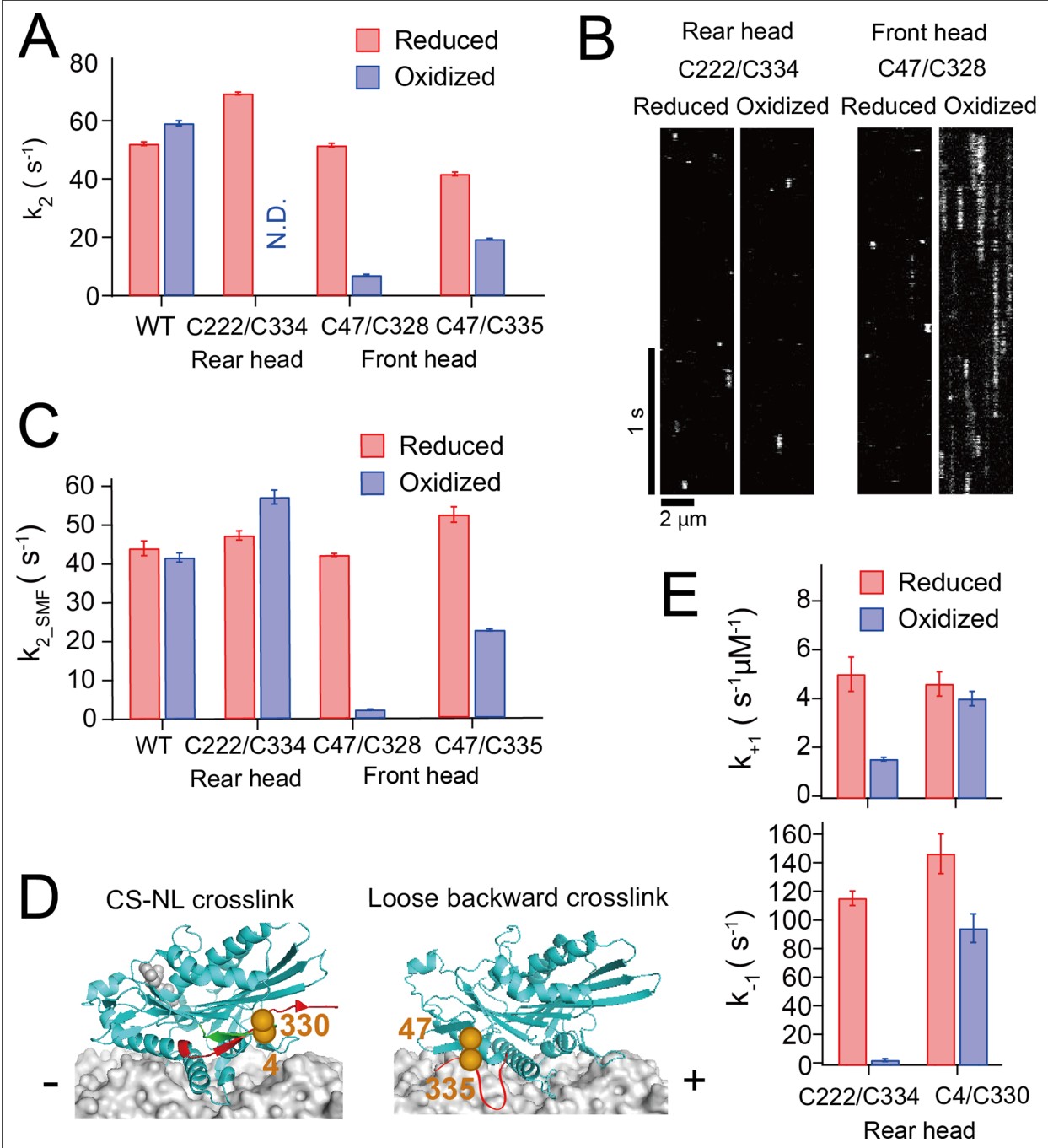

**Figure 3.** Microtubule-detachment kinetics of disulfide-crosslinked monomers. (**A**) ATP-induced dissociation rate from microtubule $k_2$ measured by turbidity change after the rapid mixing of kinesin-microtubule complex with 1 mM ATP. The turbidity time traces are shown in *Figure 3—figure supplement 1*. (**B**) Kymographs showing the binding and dissociation of GFP-fused kinesin under reduced and oxidized conditions on the microtubule. C47/C328 showed an extended dwell on the microtubule after crosslinking. (**C**) ATP-induced dissociation rate from the microtubule measured using single-molecule observations of GFP-fused kinesin. $k_{2\_SMF}$ was determined as an inverse of the mean dwell time of the fluorescent kinesin on the microtubule in the presence of 1 mM ATP. The dwell time histograms are shown in *Figure 3—figure supplement 2B*. (**D**) Positions of Cys4 and Cys330 residues for disulfide-crosslinking of a monomer between the N-terminal cover strand (CS; green) and the neck linker (NL; red) (left). Positions of Cys47 and Cys335 residues for disulfide-crosslinking of a monomer to constrain the neck linker in the backward-pointing configuration allowing partial docking of the neck linker (right; *Figure 3—figure supplement 3*). The ATP-induced dissociation rates from the microtubule of C47/C335 monomer before and after crosslinking, measured by turbidity change ($k_2$) and single-molecule fluorescence ($k_{2\_SMF}$), are shown in the figure (**A**, **C**). Histograms are displayed in *Figure 3—figure supplement 4E*. (**E**) Mant-ATP-binding rate $k_{+1}$ and dissociation rate $k_{-1}$ of C4/C330 monomer before and after crosslinking (data for C222/C334 are included from *Figure 2B* for comparison). The $k_{obs}$ plots are shown in *Figure 3—figure supplement 4B*.

*Figure 3 continued on next page*

*Figure 3 continued*

The online version of this article includes the following source data and figure supplement(s) for figure 3:

**Figure supplement 1.** ATP-induced dissociation kinetics of crosslinked kinesin from microtubule measured using light scattering.

**Figure supplement 1—source data 1.** Excel file containing light scattering trace data for *Figure 3—figure supplement 1* and *Figure 3—figure supplement 4C*.

**Figure supplement 2.** ATP-dependent microtubule-binding and -dissociation of crosslinked kinesins observed by single-molecule fluorescence microscopy.

**Figure supplement 2—source data 1.** Excel file containing microtubule-binding frequency data for *Figure 3—figure supplement 2A*.

**Figure supplement 2—source data 2.** Excel file containing microtubule-binding dwell time data for *Figure 3—figure supplement 2B* and *Figure 3—figure supplement 4E*.

**Figure supplement 3.** Locations of the C47/C328 and C47/C335 residues when the substituted head is in the front or rear of dimeric kinesin.

**Figure supplement 4.** Kinetics measurements of ATP binding and microtubule dissociation of C4/C330 and C47/C335 crosslinked monomeric kinesins.

above, the E236A mutant acts as a long-lived rear head, which is kinetically trapped in the neck-linker docked, pre-ATP hydrolysis state. Thus, in the E236A–WT heterodimer in the presence of ATP, one would expect the E236A head to be in the rear and the WT head to occupy the front.

To test this assumption, we used the smFRET sensor, which has been used to distinguish the configurations of two heads (*Mori et al., 2007*). We first introduced one cysteine into the residue 215 (located at the plus-end oriented tip of the head) of the E236A chain and another into the residue 43 (located at the minus-end oriented base of the head) of the WT chain and labeled with Cy3 and Cy5 fluorophores (*Figure 4—figure supplement 1*, left). The FRET efficiency of dual-labeled molecules bound to the microtubule in the presence of ATP showed a single peak at about 90%. When we introduced 43Cys into the E236A chain and 215Cys into the WT chain, however, the distribution of the FRET efficiency showed a single peak at about 10% (*Figure 4—figure supplement 1*, right). These results are consistent with the E236A–WT heterodimer stably taking a two-head-bound state, in which the WT head is in the front and the E236A head is in the rear position.

## ATP-binding kinetics of the front and rear heads of the heterodimer

First, we measured the steady-state microtubule-activated ATPase activity of the E236A–WT heterodimer. The microtubule-activated ATPase $k_{cat}$ of the E236A–WT heterodimer was 4.6 ATP/s per head (*Figure 6—figure supplement 1*), suggesting that the front WT head is capable of hydrolyzing ATP, but its catalytic activity is suppressed (cf. ~30 ATP/s per head for wild-type dimer *Tomishige and Vale, 2000*). As described later, the reduced ATPase rate results from suppressed microtubule detachment of the front WT head, while the rear E236A head is virtually unable to detach from microtubules.

Next, we measured the ATP-binding kinetics of the wild-type front head in the E236A–WT heterodimer using a stopped-flow apparatus. To selectively observe the fluorescent signal of mant-ATP bound to the wild-type front head, we quenched the fluorescent signal of mant-ATP bound to the E236A head by substituting residue H100 (located in loop 5 near the nucleotide pocket) in the E236A chain with cysteine and labeling it with Alexa 488 dye, which acts as an acceptor for mant-ATP (*Figure 4A*, upper). The measured ATP on-rate $k_{+1}$ and off-rate $k_{-1}$ were 5.3 $\mu M^{-1}$ $s^{-1}$ and 92 $s^{-1}$, respectively (*Figure 4B, C, F, Figure 4—figure supplement 2*), which were similar to those of the un-crosslinked monomer (*Figure 2C, F*). These results further support the notion that ATP-binding kinetics of the front head are not significantly affected by the backward strain posed to the neck linker.

We also measured the ATP-binding kinetics of the E236A rear head of the heterodimer. Pre-steady-state kinetic measurements using E236A–WT heterodimer with Alexa 488 modification on the wild-type head (*Figure 4A*, lower) revealed an ATP on-rate $k_{+1}$ of 4.4 $\mu M^{-1}$ $s^{-1}$ (*Figure 4B, C, Figure 4—figure supplement 2*), similar to that of un-crosslinked WT and E236A monomers (*Figure 2C*). Since the ATP off-rate $k_{-1}$ could not be accurately measured using stopped-flow ($k_{-1}$ was −3.3 $s^{-1}$), we used smFRET between a donor Cy3 on the E236A rear head (with cysteine modification at S43) and an acceptor Alexa-ATP. We expected ~80% high FRET when Alexa-ATP bound to the rear E236A head (dye distance ~2.5 nm; *Figure 4D*, upper) and low FRET when ATP bound to the front wild-type head (dye distance ~9 nm; *Figure 4D*, lower). The high FRET state showed a mean dwell time of 65.5 s, corresponding to $k_{-1}$ of 0.015 $s^{-1}$ (*Figure 4E, F, Figure 4—figure supplement 3*). This ATP off-rate

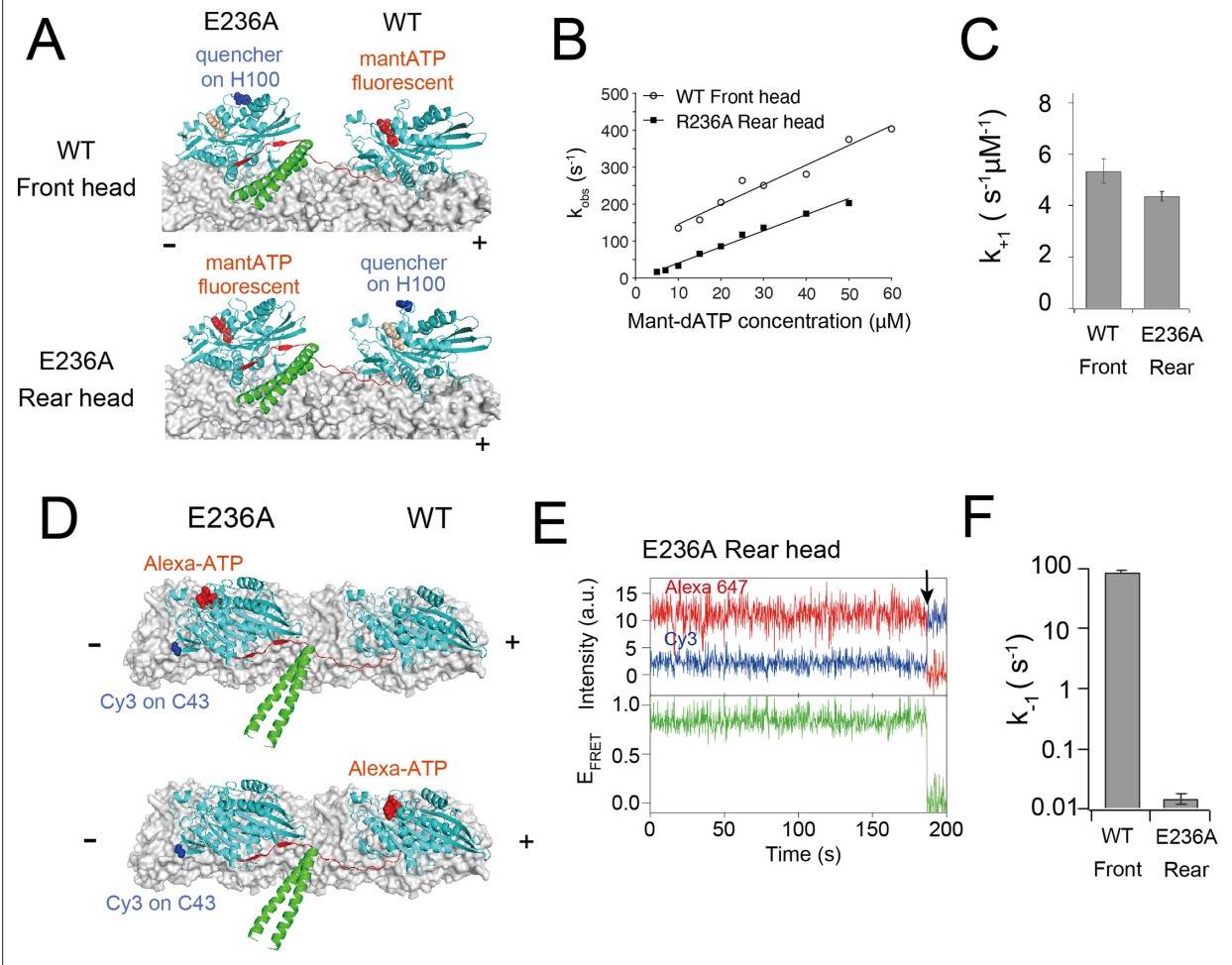

**Figure 4.** ATP-binding/dissociation kinetics of the E236A–WT heterodimer. (**A**) Diagram showing the positions of H100 residue (highlighted in blue) labeled with Alexa 488 dye. This label quenches the mant-ATP signal of either the rear W236A head (upper; used to measure ATP binding to the front head) or the front wild-type head (lower; used to measure ATP binding to the rear head). (**B**) $k_{obs}$ plots for mant-ATP binding to the front wild-type and rear E236A heads of the E236A–WT heterodimer (typical fluorescent transients are shown in *Figure 4—figure supplement 2*). Solid lines indicate a linear fit. The fit parameters are $k_{+1} = 5.3 \pm 0.5$ μM$^{-1}$ s$^{-1}$ and $k_{-1} = 92 \pm 16$ s$^{-1}$ for WT front head and $k_{+1} = 4.4 \pm 0.2$ μM$^{-1}$ s$^{-1}$ and $k_{-1} = -3.3 \pm 5.1$ s$^{-1}$ for E236A rear head. (**C**) ATP-binding rate ($k_{+1}$) for front and rear heads of the E236A–WT heterodimer, as determined by fitting in panel (**B**). (**D**) Diagram showing the position of Cys43 residue of E236A chain for labeling with donor (Cy3; blue) fluorophore, and the Alexa 647 ATP (red) bound to the rear E236A head (upper) or the front wild-type head (lower) of the E236A–WT heterodimer. High FRET efficiency is expected when the Alexa-ATP binds specifically to the rear E236A head. (**E**) A representative trace of fluorescence intensities of donor (Cy3; blue) on the E236A rear head and acceptor (Alexa 647; red) fluorophores, with calculated FRET efficiency (green), for the E236A–WT heterodimer at 200 nM Alexa-ATP recorded at 5 fps. The black arrow indicates the dissociation of Alexa-ATP, accompanied by the recovery of donor fluorescent. ATP remained bound to the rear head significantly longer compared to the E236A monomer (*Figure 2E*). (**F**) ATP dissociation rate ($k_{-1}$) for front and rear heads of the E236A–WT heterodimer. Note that these rates were measured using different methods (stopped flow for the WT font head (**B**) and single-molecule fluorescence resonance energy transfer (smFRET) for the E236A rear head (**E**)) and thus cannot be directly compared.

The online version of this article includes the following source data and figure supplement(s) for figure 4:

**Source data 1.** Excel file containing $k_{obs}$ plot data of mant-ATP binding for *Figure 4B* and *Figure 2—figure supplement 2B*.

**Figure supplement 1.** Single-molecule FRET between two heads of E236A–WT heterodimer.

**Figure supplement 1—source data 1.** Excel file containing fluorescent intensity time traces of donor and acceptor for *Figure 4—figure supplement 1B*.

**Figure supplement 2.** ATP-binding kinetics for front and rear heads of E236A–WT heterodimer.

**Figure supplement 2—source data 1.** Excel file containing mant-ATP-binding trace data for *Figure 4—figure supplement 2*.

**Figure supplement 3.** Single-molecule FRET observation between donor dye on the E236A head of E236A–WT heterodimer and acceptor-labeled ATP.

is 42-fold smaller than the E236A monomer (*Figure 2E, F*), indicating that forward strain prevents nucleotide release.

## Microtubule-detachment rate of the front head of E236A–WT heterodimer

Next, we measured the ATP-induced detachment rate $k_2$ of the front WT head of the E236A–WT heterodimer from the microtubule using high-speed single-molecule microscopy (*Figure 5A*; *Isojima et al., 2016*). We labeled the WT head with a gold nanoparticle (40 nm in diameter) and observed the motion of the gold probe attached to the E236A–WT heterodimer on the microtubule in the presence of 1 mM ATP. With the temporal resolution of the measurement (50 µs), we could clearly identify transient increases in the fluctuation of the gold probe accompanied by detachment of the labeled head (*Figure 5B*). Moreover, these observations allowed us to distinguish whether the gold-labeled WT head was in the leading or trailing position just before microtubule detachment; the backward displacement of the detached head indicates that the labeled WT head occupied the leading position prior to detachment (*Figure 5—figure supplement 1*). Unlike the unbound trailing head of wild-type dimer that showed continuous mobility (*Isojima et al., 2016*), the unbound WT head of E236A–WT heterodimer exhibited a low-fluctuation state in the middle (*Figure 5B* and s.d. traces). This low-fluctuation unbound state was distinguishable from the typical microtubule-bound state, having a shorter dwell time of ~5 ms compared to the bound state and positioning backward, closer to the E236A head, relative to the bound state (*Figure 5—figure supplement 2*). The mean dwell time of the bound and unbound states of the front head at 1 mM ATP was 160 and 11 ms, respectively (*Figure 5C* and *Figure 5—figure supplement 3*).

To determine the rate constants for the ATP-promoted detachment rate of the front WT head, we measured the dwell time of the microtubule-bound state of the gold-labeled WT head under various ATP conditions (*Figure 5D* and *Figure 5—figure supplement 3*). The mean dwell time increased as ATP concentration was decreased (*Figure 5E*), indicating that the bound dwell time includes ATP binding and hydrolysis in the front head. The inverse of the mean dwell time plotted against ATP concentration could be fit well with a Michaelis–Menten equation with fit parameters of $k_{cat}$ = 6.3 ± 0.2 s⁻¹ and $K_m$ = 43 ± 6 µM (*Figure 6D*). The observed $k_{cat}$ also includes the ADP release rate; however, since ADP release (~5 ms) becomes negligible within the dwell time of >160 ms, we took this $k_{cat}$ as an approximation of the ATP-promoted detachment rate $k_2$. The $k_2$ determined using E236A–WT heterodimer (6.3 s⁻¹) is similar to the $k_2$ of front head crosslink (7.0 s⁻¹), supporting the notion that the backward strain posed to the neck linker significantly reduces nucleotide-induced detachment rate of the front head. In contrast, the dwell time of the unbound state of the gold-labeled WT head showed weak ATP dependence (*Figure 5—figure supplement 2*), indicating that the rear E236A head occasionally releases ATP when the front head detaches from the microtubule and the neck linker of E236A head becomes unconstrained. This finding further supports the idea that forward neck-linker strain plays a crucial role in reducing the reversible ATP release rate.

## Effect of the neck-linker tension on the gating of the front and rear heads

Experimental results obtained from neck-linker crosslinking monomers and the E236A–WT heterodimer revealed that distinct chemical steps are gated in the front and rear heads ($k_2$ and $k_{-1}$, respectively). We next sought to examine whether neck-linker tension plays a role in gating these steps. To reduce the tension posed to the neck linker in the two-head-bound state, we extended the neck linker of the E236A–WT heterodimer by inserting 7 or 12 glycine residues between the neck linker and neck coiled-coil of both chains (referred to as G7 and G12) (*Figure 6A*; *Isojima et al., 2016*). The microtubule-activated ATPase rate of G12 increased modestly by 1.6-fold compared to that of the heterodimer without neck-linker extension (referred to as G0) (*Figure 6B* and *Figure 6—figure supplement 1*).

We first examined the effect of the tension posed to the neck linker on ATP-induced detachment rate $k_2$ of the front head by observing the neck-linker extended E236A–WT heterodimer using high-speed dark-field microscopy. The gold probe attached to the front WT head of G7 and G12 heterodimers showed transient increases in the fluctuation of the gold probe (*Figure 6C*). The inverse of the dwell time of the bound state for G7 and G12 mutants (*Figure 6—figure supplements 2*

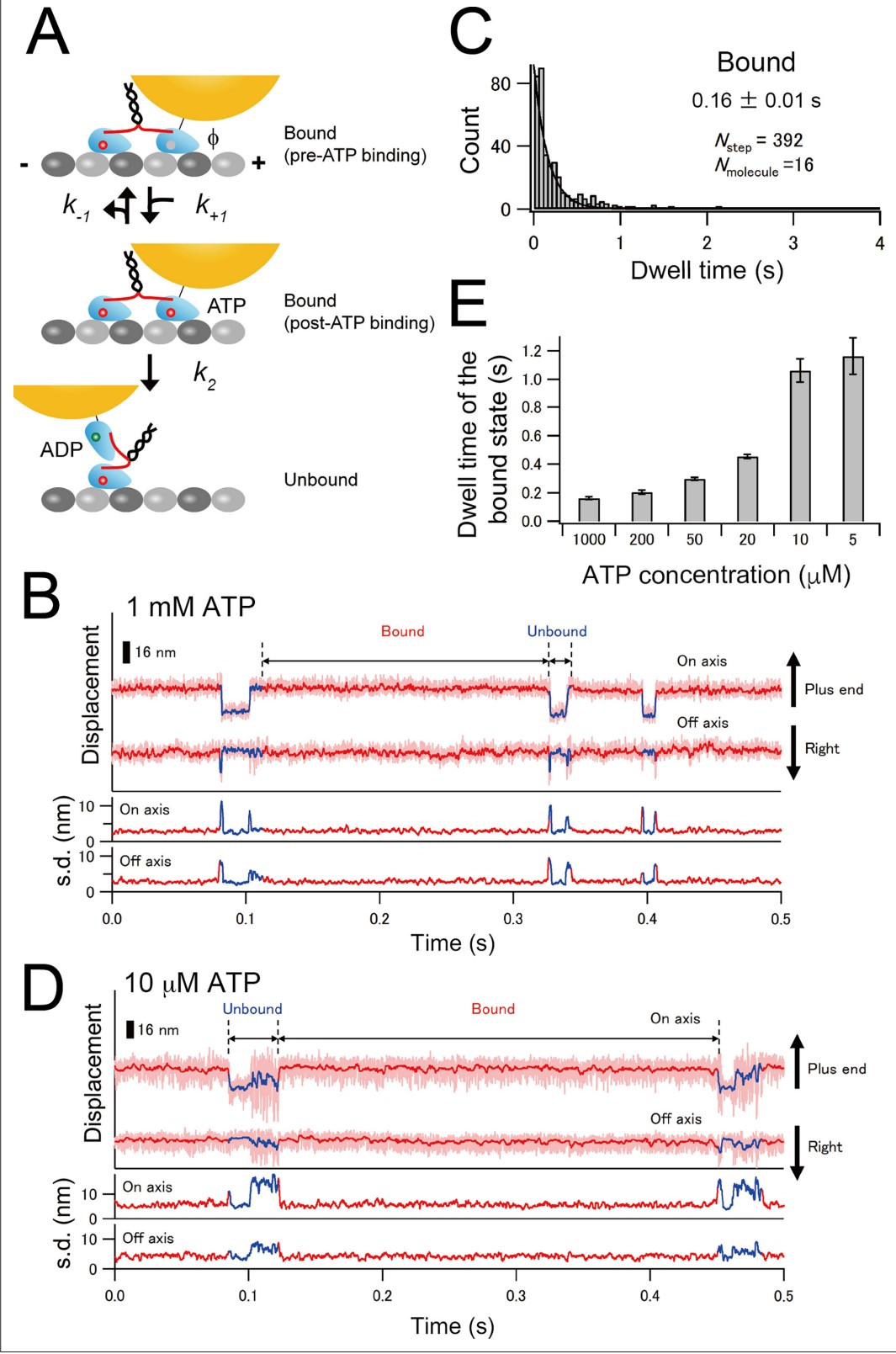

**Figure 5.** Microtubule-detachment kinetics of the front WT head of E236A–WT heterodimer. (**A**) A diagram illustrating the kinetic transitions for the detachment of the gold-labeled WT head of the E236A–WT heterodimer. Upon binding to the microtubule, the front head releases ADP, becoming a nucleotide-free state (indicated as ϕ; pre-ATP binding). The detachment kinetics from this state can be described using the same scheme as the

*Figure 5 continued on next page*

*Figure 5 continued*

monomer (*Figure 1B*). (**B**) Typical trace for the centroid positions of the gold probe attached to the WT head of the E236A–WT heterodimer in the presence of 1 mM ATP (light red lines), toward the microtubule long axis (on axis) and perpendicular to the microtubule axis (off axis). Red and blue lines depict the median-filtered trace (with a window size of 51 frames) for the bound and unbound states, respectively. Lower panels display the standard deviation (s.d.) of on- and off-axis positions for each time frame $t$ (calculated as $[t – 20, t + 20]$). (**C**) Histogram of the dwell time in the bound state. The solid line shows the fit with an exponential function. The number represents the average dwell time (± SEM) determined from the fit. (**D**) Typical trace for the centroid positions of the gold probe attached to the WT head of E236A–WT heterodimer in the presence of 10 µM ATP. (**E**) Mean dwell times in the bound state under various ATP concentrations as determined from the fit of the histograms of the dwell times (*Figure 5—figure supplement 3*).

The online version of this article includes the following source data and figure supplement(s) for figure 5:

**Source data 1.** Excel file containing 2D time traces of centroid position of gold probe for *Figures 5B, D and 6C*, *Figure 5—figure supplement 1*, and *Figure 6—figure supplement 4A*.

**Figure supplement 1.** Long-term trace of the gold-labeled E236A–WT heterodimer exhibiting very slow stepping motion.

**Figure supplement 2.** The unbound state of the gold-labeled WT head of the E236A–WT heterodimer.

**Figure supplement 2—source data 1.** Excel file containing dwell time histogram data of unbound state for *Figure 5—figure supplement 2B*.

**Figure supplement 2—source data 2.** Excel file containing 2D histogram data of unbound state for *Figure 5—figure supplement 2C*.

**Figure supplement 3.** Distributions of the dwell time in the bound and unbound states of the leading WT head of E236A–WT heterodimer.

**Figure supplement 3—source data 1.** Excel file containing dwell time histogram data of E236A–WT heterodimer for *Figure 5—figure supplement 3* and *Figure 6—figure supplement 2*, *Figure 6—figure supplement 3* and *Figure 6—figure supplement 4C*.

---

*and 3*) plotted against ATP concentration fit well with the Michaelis–Menten equation (*Figure 6D*); the fit parameters are $k_{cat}$ = 7.6 ± 0.9 s$^{-1}$ and $K_m$ = 22 ± 10 µM for G7 and $k_{cat}$ = 9.3 ± 0.7 s$^{-1}$ and $K_m$ = 6.1 ± 2.1 µM for G12. The $k_2$ modestly increased as the inserted poly-Gly number increased; G12 showed a 1.5-fold increase in $k_2$ compared to G0 (*Figure 6E*), reminiscent of the results of ATPase rates (*Figure 6B*). The detached WT head of the G12 heterodimer occasionally bound to the rear-tubulin-binding site (the frequency was 14% among all binding events); however, the head detached from the rear-binding position much faster (19 ms at 1 mM ATP; *Figure 6—figure supplement 4*) than from the front-binding site (111 ms), indicating that the orientation of the neck linker has a greater effect on the front-head gating than the tension. These findings demonstrate that the detachment rate of the front head is not significantly affected by the amount of tension posed to the neck linker, at least within the range of 12 glycine insertions.

Next, we examined the effect of the tension posed to the neck linker on ATP off-rate $k_{-1}$ of the rear E236A head of the E236A–WT heterodimer by measuring smFRET between Cy3-labeled E236A head and Alexa-ATP. When compared to the heterodimer without neck-linker extension (0.015 s$^{-1}$; *Figure 4E*), $k_{-1}$ determined from the inverse of the mean dwell time of the high FRET state only modestly increased for G7 (0.018 s$^{-1}$) and increased by fivefold for G12 (0.081 s$^{-1}$) (*Figure 6F* and *Figure 6—figure supplement 5*). The $k_{-1}$ was further increased for monomeric E236A without neck-linker constraint as described before (0.65 s$^{-1}$; *Figures 2F and 6F*). These results indicate that the ATP off-rate of the rear E236A head is significantly affected by the amount of the forward strain, especially when the neck linker was extended with more than 7 glycine insertions.

## Discussion

To identify the mechanochemical transition that is gated in the front and/or rear head, we measured ATP binding and microtubule-detachment kinetics of kinesin's catalytic domain with neck-linker constraints imposed by (1) disulfide-crosslinking of the neck linker in monomeric kinesin in a backward (front head-like) and forward (rear head-like) orientations, and (2) using the E236A–WT heterodimer to create a two-head microtubule-bound state with the mutant and WT heads occupying the rear and

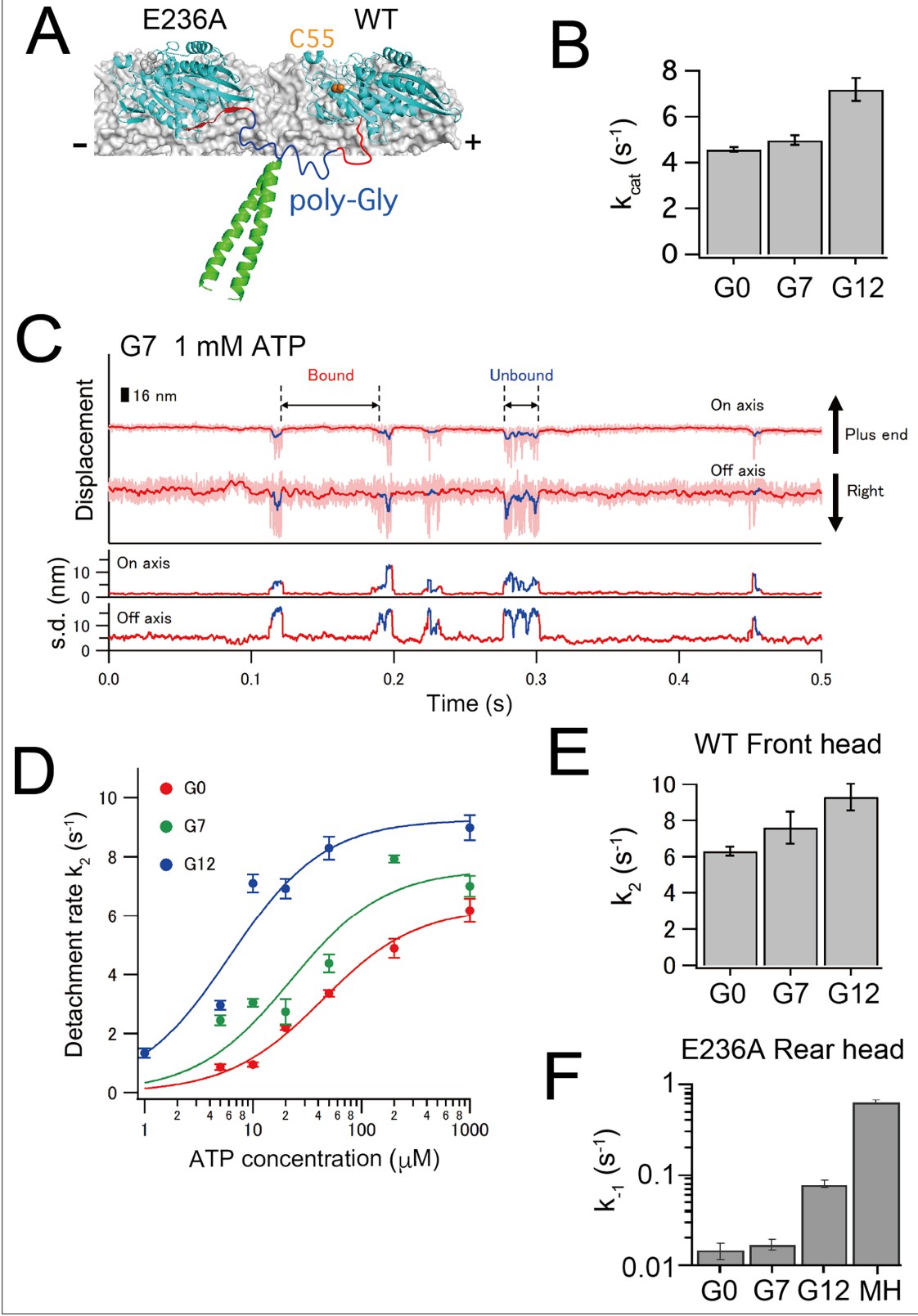

**Figure 6.** Effect of the tension applied to the neck linker on the gating examined using the neck-linker extended E236A–WT heterodimer. (**A**) Diagram showing the position of the poly-Gly residues (blue) inserted between the neck linker (red) and the neck coiled-coil (green). The Cys55 residue for gold labeling is represented by an orange sphere. (**B**) $k_{cat}$ for the microtubule-activated ATPase of E236A–WT heterodimers without (termed as G0) and with neck-linker extensions (G7 and G12) (**Figure 6—figure supplement 1**). (**C**) Typical trace for the centroid positions of the gold probe attached to the WT

*Figure 6 continued on next page*

*Figure 6 continued*

head of the E236A–WT heterodimer with a 7 poly-Gly insertion in the presence of 1 mM ATP. (**D**) The inverse of the mean dwell time in the bound state was plotted as a function of ATP concentration, as determined from the fit of the histograms of the dwell times (*Figure 6—figure supplements 2 and 3*). Error bars represent SEM. The solid lines show the fit with the Michaelis–Menten equation. The fit parameters are $k_{cat}$ (or $k_2$) = 6.3 ± 0.2 s$^{-1}$ and $K_m$ (ATP) = 43 ± 6 µM for G0, $k_{cat}$ = 7.6 ± 0.9 s$^{-1}$ and $K_m$ (ATP) = 22 ± 10 µM for G7, and $k_{cat}$ = 9.3 ± 0.7 s$^{-1}$ and $K_m$ (ATP) = 6.1 ± 2.1 µM for G12. The $K_m$ (ATP) decreases as the insertion length increases. (**E**) The $k_{cat}$ value, which represents the ATP-induced detachment rate of the front head $k_2$, obtained from the fit shown in panel D. The error bars represent SEM. (**F**) The ATP dissociation rates $k_{-1}$ for the E236A rear head of the wild-type (G0), G7, and G12 E236A–WT heterodimer were determined using single-molecule fluorescence resonance energy transfer (smFRET). The $k_{-1}$ was calculated as the inverse of the mean dwell time for the high FRET state (see the typical trace and dwell time histograms for the G7 and G12 heterodimers in *Figure 6—figure supplement 5*). For comparison, data for the E236A monomer head (referred to as MH) is included from *Figure 2F*.

The online version of this article includes the following source data and figure supplement(s) for figure 6:

**Source data 1.** Excel file containing microtubule-detachment rate data for *Figure 6D*.

**Figure supplement 1.** Microtubule-activated ATPase rates of the E236A–WT heterodimer with and without neck-linker extension.

**Figure supplement 1—source data 1.** Excel file containing ATPase measurement data for *Figure 6—figure supplement 1*.

**Figure supplement 2.** Distributions of the dwell time in the bound and unbound states of the leading WT head of E236A–WT heterodimer with 7 poly-Gly insertion (G7).

**Figure supplement 3.** Distributions of the dwell time in the bound and unbound states of the leading WT head of E236A–WT heterodimer with 12 poly-Gly insertion (G12).

**Figure supplement 4.** Binding to the rear-tubulin-binding site observed for the G12 E236–WT heterodimer.

**Figure supplement 5.** Single-molecule FRET between donor-labeled E236A head of E236A–WT heterodimer with extended neck linker and acceptor-conjugated ATP.

**Figure supplement 5—source data 1.** Excel file containing fluorescent intensity time traces of donor and acceptor for *Figure 6—figure supplement 5A*.

front positions, respectively. In addition to the orientational constraints on the neck liner, we examined the effect of its tension using neck-linker extended mutants of the E236A–WT heterodimer. As also described in the Results, these two experimental paradigms each have unique caveats in terms of how faithfully they replicate a native moving kinesin dimer. We also used potentially perturbing measurement approaches that involve using fluorescently labeled ATP or modifying cysteine-light kinesin with fluorescent dyes or gold particles. However, the results from these two different experimental approaches, along with the variety of different measurement techniques (stopped flow, smFRET, high-speed single-molecule microscopy), consistently demonstrate a relationship between directional strain on the neck linker and its effect on the catalytic state of the motor domain, as summarized below.

In the front head, the backward-pointing orientation of the neck linker has little effect on ATP binding and dissociation rates, both when measured for a monomer crosslink (*Figure 2A, B*) and for the front head of an E236A–WT heterodimer (*Figure 4B, C, F*). These findings differ from a prevailing view that the ATP-binding step is gated in the front head (*Rosenfeld et al., 2003*; *Klumpp et al., 2004*; *Guydosh and Block, 2006*; *Dogan et al., 2015*). However, we found that the ATP-induced detachment rates from microtubule ($k_2$) were similarly reduced for both the front head crosslink (7.0 s$^{-1}$; *Figure 3A*) and the front WT head of the E236A/WT heterodimer (6.3 s$^{-1}$; *Figure 6D*), suggesting that a step subsequent to ATP binding is gated in the front head. These results are consistent with an inability of the front head to fully close its nucleotide pocket to promote ATP hydrolysis and Pi release (*Benoit et al., 2023*), as will be discussed later.

We also provide evidence for a rear-head gating mechanism that involves a tighter affinity for ATP (due to a decrease in reversible ATP dissociation rate $k_{-1}$) without a change in microtubule affinity in the ATP-bound state. The $K_d$ of ATP for the front head crosslink and the monomer without constraint are similar (~20 µM) and substantially higher than our estimated 4 nM $K_d$ of ATP for the rear E236A head of the E236A–WT heterodimer. We note, however, that this $K_d$ of ATP may somewhat underestimate the actual value in wild-type kinesin for two reasons: first, the E236A mutation likely stabilizes the neck linker docked, closed state more than in the rear head of the wild-type dimer (*Rice et al., 1999*), and second, the Alexa-ATP used to measure the ATP off-rate of E236A head showed ~10-fold smaller velocity compared to unmodified ATP, partly due to a slower ATP off-rate (*Figure 2—figure supplement 3*). While these caveats may diminish the observed 5000-fold difference in the $K_d$ of ATP in native kinesin, these results nevertheless point to the large difference in ATP affinity between the

front and rear heads. The difference in the ATP-affinity between the front and rear heads can explain previous experimental results supporting the front-head gating mechanism (*Uemura and Ishiwata, 2003*; *Rosenfeld et al., 2003*; *Klumpp et al., 2004*; *Guydosh and Block, 2006*; *Dogan et al., 2015*), although our direct kinetic measurements show that this difference is manifest through in a stronger affinity for ATP in the rear head rather than a weaker affinity in the front head. While the neck-linker position of the rear head increases the affinity of bound ATP, it does not significantly accelerate $k_2$, which includes the ATP hydrolysis and Pi release steps in the nucleotide catalysis cycle.

## Kinetic gating aids head–head coordination but does not prevent progression through the ATPase cycle

Our results show that the front-head gate slows down but does not block its ATPase cycle. The ATP-induced microtubule-detachment rate $k_2$ of the front head is 6–7 s$^{-1}$ (7.0 s$^{-1}$ for the front head monomer crosslink and 6.3 s$^{-1}$ for the front head of E236A–WT heterodimer), which is similar to its overall ATPase rate (4.6 s$^{-1}$). These rate constants are sufficiently low to keep the catalytic cycles of two heads largely out of phase, since the detachment rate of the rear head of wild-type dimer (inverse of the dwell time from two-head- to one-head-bound state during processive movement) is ~100 s$^{-1}$ (*Isojima et al., 2016*). The observed detachment of the front head of the E236A–WT heterodimer is likely caused by ATP hydrolysis of the front head, because (1) the ATP turnover rate of this heterodimer was similar to the $k_2$ value, and (2) the dwell time of the bound state of the front head on microtubule was dependent on the ATP concentration. These results suggest that the kinesin's head can slowly hydrolyze ATP and detach from the microtubule without the aid of neck-linker docking. This idea can explain how the kinesin dimer takes repetitive back steps under large hindering loads. *Carter and Cross, 2005* showed that kinesin takes 8 nm backward steps at supra-stall forces (>7 pN) exerted using an optical trap and that the stepping rate was dependent on the ATP concentration (~0.3 and ~1 step/s at 1 mM and 10 µM ATP conditions, respectively), which are comparable to the dwell times that we found for the microtubule-bound state of the front head of E236A–WT heterodimer (0.16 and 1.1 s at 1 mM and 10 µM ATP, respectively; *Figure 5—figure supplement 3*). These results suggest that the back steps or slips are likely caused by the neck-linker docking-independent slow ATP catalysis of the front head, followed by its dissociation from the microtubule in an ADP-bound state.

## A kinetic and structural model for kinesin gating

The difference in the ATP-binding affinity between the front (low affinity) and rear (high affinity) heads suggests a structural difference in their nucleotide binding pockets. Indeed, the high ATP affinity state of the neck linker docked rear head is consistent with a 'closed' nucleotide pocket observed in crystal and cryo-EM structures of kinesin-1 bound to tubulin and complexed with a non-hydrolyzable ATP analog in kinesin's active site (*Shang et al., 2014*; *Gigant et al., 2013*). Moreover, a recent high-resolution cryo-EM image of the microtubule-bound KIF14 dimer showed that while they could observe nucleotide in the active site of the front head, the structure of the pocket was more 'open' and similar to that of a nucleotide-free head (*Shang et al., 2014*; *Cao et al., 2014*; *Benoit et al., 2021*). Since the switch I and II loops that are involved in the hydrolysis reaction are positioned away from the γ-phosphate in the opened nucleotide pocket, the open state is incompatible with ATP hydrolysis, while the closed state is competent in ATP hydrolysis (*Kull and Endow, 2002*; *Parke et al., 2010*; *Figure 7—figure supplement 1B*). Thus, these structural features are consistent with our kinetics results.

These findings suggest that the isomerization from the open to closed conformational state, rather than the transitions between nucleotide states, is gated by the neck-linker constraint (*Figure 7—figure supplement 2*). This aligns with our structural modeling of the kinesin dimer in the two-head-bound state (*Makino et al., 2024*), which showed that conformational transitions are suppressed for both the front head (from open to closed state) and the rear head (from closed to open state), as these changes would cause an intolerable increase in tension on the disordered neck-linker region (*Figure 7*). We demonstrated, using the loose C47/C335 backward crosslink of the neck linker, that neck-linker docking is responsible for opening the gate (i.e., the open-to-closed transition) (*Figure 3A–D*). The C47/C335 crosslink forms a short 7-amino acid flexible loop, which is insufficient for the formation of two β-sheets (residues 326–334) (*Figure 3—figure supplement 3*). However, it allows the conformational transition of the initial segment of the neck linker (residues 323–325); K323 and T324 residues

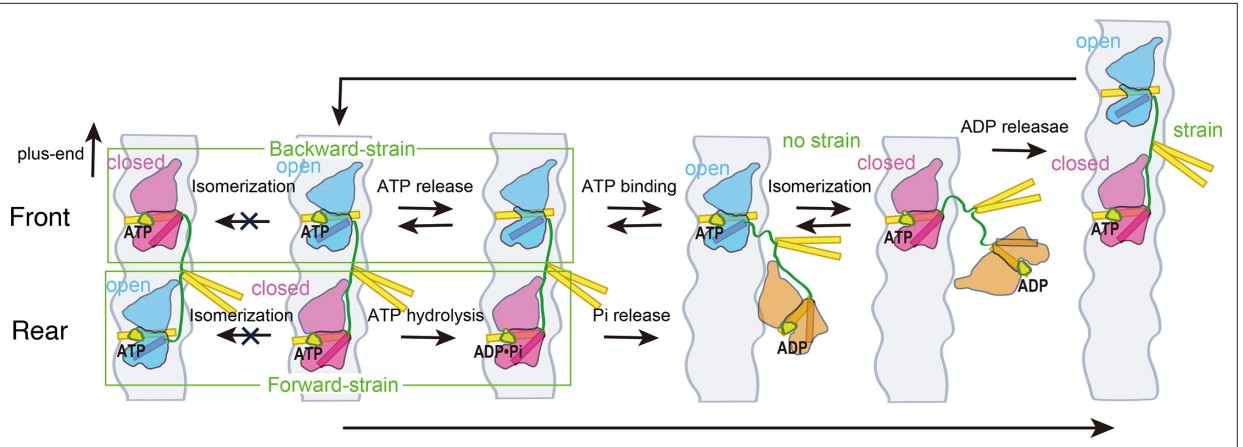

**Figure 7.** Model to explain how the tension exerted by the stretched neck linker in dimeric kinesin coordinates the microtubule detachment of the two heads. This model is based on the proposal that the transition between ATP-bound open and closed conformations of the head is regulated by the neck-linker strain (*Figure 7—figure supplement 2*). The open and closed conformational states of the head are indicated in blue and red, respectively, with the α6 helix, which connects to the neck linker, highlighted as a rod. The microtubule-detached ADP-bound state is shown in orange. The neck linker is depicted in green, while the α4 helix, which directly interacts with the microtubule, and the neck coiled-coil are shown in yellow. In the two-head-bound state, the front head remains in the open state because the backward strain prevents it from transitioning to the closed state. In this state, ATP can weakly bind to the nucleotide pocket but often dissociates. Conversely, the rear head is stabilized in the closed state because the transition to the open state is suppressed by the forward strain. ATP is tightly bound to the closed nucleotide pocket, causing the rear head to hydrolyze ATP and detach from the microtubule before the front head does. In the one-head-bound state, the microtubule-bound head can transition between the open and closed states. However, once the tethered head binds preferentially to the forward tubulin-binding site, the strain built up between the two microtubule-bound heads stabilizes the rear head in the closed conformation and the front head in the open conformation.

The online version of this article includes the following figure supplement(s) for figure 7:

**Figure supplement 1.** Structural difference between the open and closed conformational states of the kinesin head.

**Figure supplement 2.** Schematic model showing that the open-closed conformational transition (isomerization) is gated by the neck-linker strain.

form an α-helix extending the C-terminus of α6, while the I325 residue plays a particularly important role in docking onto the hydrophobic pocket exposed on the closed head. Given that the hydrophobic pocket is positioned forward from the neck linker's base (*Figure 7—figure supplement 1A*), the docking of the I325 side chain is suppressed in the front head, as it would cause an intolerable increase in tension (entropy decrease) on the disordered neck linker (*Makino et al., 2024*), thus maintaining the open state. The neck-linker extension had a smaller effect on front-head gating (*Figure 6E*), likely because a stable interaction between the I325 side chain and the hydrophobic pocket is necessary to maintain the closed state until ATP hydrolysis completes. In contrast, the neck-linker extension had a larger effect on the rear head's ATP affinity (*Figure 6F*). This is probably because the rear head can occasionally transition back to an open state (followed by reversible ATP release) without tension on the neck linker (*Figure 2D*; *Rice et al., 2003*). Thus, forward strain plays a crucial role in aiding I325 residue docking and stabilizing the closed state.

In summary, our findings, together with the structural data discussed above, are consistent with an asynchronous sequence of events in the front and rear kinesin heads that help to coordinate their chemomechanical cycles to ensure their continuous stepping along the microtubule (*Figure 7*). In the rear head, ATP binding is followed rapidly by an isomerization step that closes the nucleotide pocket, decreasing the rate of nucleotide dissociation and facilitating ATP hydrolysis. This conformational change is facilitated by the docking of the neck linker in its forward orientation (*Benoit et al., 2023*). In the front head, in contrast, the backward conformation of the neck linker favors an open nucleotide pocket, which helps to prevent the microtubule-bound front head from proceeding with ATP hydrolysis before the rear head detaches. Once the rear head detaches, the front head 'gate' is relieved, allowing the neck linker to dock and its nucleotide pocket to close. This asynchrony in the timing of ATP hydrolysis minimizes the likelihood of both heads simultaneously entering an ADP state and dissociating from the microtubule. Recent cryo-EM studies and modeling of the kinesin dimer under various nucleotide states support the proposal that the open-to-closed conformational transition is

gated by the neck-linker strain; however, direct evidence for this conformational transition in the context of a processively moving kinesin dimer would help to further substantiate this model.

## Materials and methods
### DNA cloning and protein purification
We used a 339 amino acid 'cysteine-light' mutant or a 349 amino acid wild-type human kinesin-1 monomer as a template for mutagenesis. Cysteines and/or a E236A point mutation were introduced by PCR cloning involving QuikChange mutagenesis (Stratagene) (*Tomishige and Vale, 2000*). For heterodimer, mutations were introduced into one of the two polypeptide chains of the cysteine-light kinesin-1 heterodimer with 490 amino acids (*Tomishige et al., 2006*; *Isojima et al., 2016*). For single-molecule fluorescent observation of monomers, GFP (P64L/S65T variant) was fused at the C-terminal of constructs. All the constructs were verified by DNA sequencing. Kinesin proteins were expressed and purified as described previously (*Tomishige and Vale, 2000*; *Tomishige et al., 2006*) except that for monomeric kinesins, Ni-NTA chromatography was followed by dialysis against 25 mM PIPES (pH 7.0), 100 mM NaCl, 2 mM MgCl$_2$, and 100 µM ATP for 3 hr at 4°C. EGTA in the buffer was removed. For steady-state ATPase assays and single-molecule observations, proteins were further purified with microtubule-affinity purification (*Tomishige et al., 2006*). DTT and EGTA were removed from the solutions. Tubulin was purified and polymerized as described previously (*Woehlke et al., 1997*). Protein concentrations were determined by Bradford assays using BSA as a standard. For fluorescent labeling (smFRET or mant-ATP-binding measurement of heterodimer), dialyzed kinesin was reacted with Cy3-maleimide (Cytiva), Cy5-maleimide (Cytiva), ATTO488-maleimide (ATTO-TEC), or Alexa488-maleimide (Thermo Fisher). The labeled protein was further purified through microtubule-affinity purification as described (*Yildiz et al., 2004*; *Tomishige et al., 2006*).

### Chemical crosslinking
Disulfide crosslink was formed by oxidation using copper and phenanthroline (*Kobashi, 1968*; *Kaan et al., 2011*). Reactions were carried out by addition of 10 µM CuCl$_2$ and 20 µM *o*-phenanthroline to 4 µM kinesin heads in 25 mM PIPES (pH 6.8) plus 2 mM MgCl$_2$ and incubation on ice for 3 hr. The reactions were quenched by addition of 1 mM EGTA. Control reactions were carried out in parallel by adding 10 mM DTT. Disulfide-crosslinking was verified using non-reducing SDS–PAGE with AMS (4′-acetamido-4′-maleimidylstilbene-2,2′-disulfonic acid, 6508; Setareh Biotech) which react with reduced cysteine residues causing band shift on SDS–PAGE (*Denoncin et al., 2013*). Post crosslinking reaction of kinesin (after addition of EGTA) was denatured by addition of 10% trichloroacetic acid and incubation for 30 min at 4°C. 100 µl of denatured kinesin solution was centrifuged (100k rpm, for 5 min) and pellet was carefully washed with 50 mM Tris-HCl (pH 7.5) plus 10 mM EDTA to remove DTT or copper. The pellet was then resuspended in 30 µl of solution containing 50 mM Tris-HCl (pH 7.5), 10 mM EDTA, 0.1% SDS, 20 mM AMS. The sample was incubated over night at room temperature under shaking. Addition of SDS sample buffer without reducing reagents was followed by SDS–PAGE (8.5% polyacrylamide) and CBB staining. Ratio of reduced and oxidized molecules was quantified using ImageJ.

### Steady-state ATPase assay
Microtubule-stimulated ATPase activity was measured using spectrometer (V550; JACSO) with a coupled enzymatic assay as described previously (*Woehlke et al., 1997*). The ATPase assays were performed in BRB12 buffer (12 mM PIPES (pH 6.8), 2 mM MgCl$_2$, 1 mM EGTA) containing 1 mM ATP, 20 µM taxol, 0.1 mg/ml casein, and coupled NADH oxidation system (0.2 mM NADH, 5 mM phospho(enol)pyruvate, 10 µg/ml of pyrvate kinase, and 10 µg/ml of lactate dehydrogenase). 1 mM DTT was also added when measuring reducing conditions. The assays were started with 10 nM kinesin and varying concentrations of microtubules at 22°C.

### Pre-steady-state kinetics measurements
All stopped-flow experiments were performed at 25°C in BRB25 buffer (25 mM PIPES (pH 6.8), 2 mM MgCl$_2$, 1 mM EGTA) using a KinTek stopped-flow system (SF2004; Kintek Corp, State College, PA) with a Xenon lamp. Mant-nucleotide fluorescence measurements were performed using an extension

wavelength of 280 nm and fluorescence emission was monitored using a 445/40 band pass filter (D445/40M; Chroma Tec). Five to ten fluorescent traces were averaged for each experimental condition. Turbidity measurements were performed at 340 nm and 15–20 turbidity traces were averaged. Averaged stopped-flow traces were fit to single exponential function,

$$y = A \, exp\left(-k_{obs}t\right) + C \tag{1}$$

or a burst equation,

$$y = A \, exp\left(-k_{obs}t\right) + k_{ss}t + C \tag{2}$$

where $k_{obs}$ is the rate constant of the initial exponential phase, $k_{ss}$ is the rate constant of the linear phase, $A$ is the amplitude of exponential phase and $C$ is the constant term.

### Mant-ATP-binding kinetics

Microtubule–kinesin complex was prepared by incubation of 3.2 µM post crosslinking kinesin with 4 µM microtubule and 20 µM taxol in BRB25 buffer at room temperature for 10 min. Microtubule-C222/C334 kinesin complex was prepared by following procedure. 3.2 µM of reduced/oxidized C222/C334 kinesin was incubated with 4 µM microtubules, 20 µM taxol, and 5 U/ml apyrase at room temperature for 1 hr, followed by a centrifugation (80k rpm, 10 min) on 20% sucrose cushion. For Alexa488-labeled heterodimer, 5 µM of heterodimer kinesin was incubated with 25 µM microtubules in the presence of 1 mM ATP at room temperature for 10 min, followed by a centrifugation (80k rpm, 10 min) on 60% glycerol cushion. The pellet was resuspended in initial volume of BRB25 plus 20 µM taxol. The concentration of the heterodimer–microtubule complex was estimated using the extinction coefficient of Alexa488 dye (73,000 cm$^{-1}$ M$^{-1}$) and was adjusted to approximately 1 µM after mixing. The microtubule–kinesin complex was then rapidly mixed with increasing concentrations of mant-ATP (M12417; Invitrogen) or mant-dATP (3′-O-(N-methyl-anthraniloyl)-2′-deoxyadenosine-5′-triphosphate; Jena Bioscience). Fluorescence data of mantATP binding were fit to single exponential (*Equation 1*). Values of $k_{obs}$ were plotted against nucleotide concentration and fit to linear function,

$$k_{obs} = k_{+1} \left[\text{mantATP}\right] + k_{-1} \tag{3}$$

where $k_{+1}$ is the second-order rate constant for nucleotide binding and $k_{-1}$ is the rate constant for nucleotide dissociation from the motor.

### Microtubule dissociation kinetics

Microtubule–kinesin complex was prepared by incubation of 3.2 µM post crosslinking kinesin with 4 µM microtubule, 20 µM taxol, and 0.1 U/ml apyrase in BRB25 buffer at room temperature for 10 min. 150 mM KCl was then added into kinesin solution. The change in turbidity was monitored after rapid mixing of microtubule–kinesin complex with ATP (1 mM post mixing) plus 150 mM KCl. Turbidity traces were fit to burst equation (*Equation 2*) to determine microtubule dissociation rate.

### Single-molecule fluorescence observation of GFP-labeled kinesin

Binding/detachment of individual GFP-fused kinesin on axonemes (purified from sea urchin sperm flagella) were observed in a custom-build prism-type laser-illuminated total-internal-reflection microscope as described (*Tomishige and Vale, 2000*). Assay chamber constructed between a quartz slide and a coverslip was first filled with axonemes diluted in BRB12 for 3 min, followed by a washing with 1 mg/ml casein in BRB12 for 3 min. The chamber was then filled with the assay solution contained GFP-kinesin (200 pM for WT, C47/C328, and C47/C335, 500 pM for C222/C334 and C4/C330), 1 mM ATP, 100 mM KCl, and oxygen scavenging system (4.5 mg/ml glucose, 50 U/ml catalase, 50 U/ml glucose-oxidase) and sealed. 70 mM 2-mercaptoethanol was added in the assay solution when observing in reducing condition. GFP molecules were exited at 488 nm and fluorescence was recorded at 10 ms temporal resolution. Distributions of dwell time on axonemes were obtained from kymographs. Fluorescent spots dwelled more than two frames were used for analysis. Detachment rates were determined by single exponential fitting of the dwell histograms.

### Flow-cell chamber preparation for single-molecule FRET

For the wild-type monomeric kinesin, flow-cell chamber was constructed between poly-L-lysine-coated quartz slide and cover slip. The chamber was first filled with microtubule in BRB12 buffer containing 20 µM taxol for 5 min. Then, BRB12 buffer containing 20 µM taxol and 1 mg/ml casein was infused and incubated for 2 min followed by washing with BRB12 buffer containing 20 µM taxol. The chamber was finally filled with observation buffer containing an ATP regenerating system (10 µg/ml creatin kinase, 2 mM creatin phosphate), an oxygen scavenging system (2.5 mM protocatechuic acid, 1 unit/ml protocatechuate-3,4-dioxygenase (P8279-25UN; Sigma), 2 mM Trolox) (*Aitken et al., 2008*), 150 mM KCl, 20 µM taxol, 500 nM Alexa Fluor 647 ATP (A22362; Invitrogen), and 100 pM kinesin in BRB12 buffer. All procedures were carried out at room temperature.

For the other constructs (E236A monomer, E236A–WT heterodimer), flow-cell chamber was constructed between quartz slide pre-cleaned with 0.1 M KOH and cover slip, separated by ~50 µm thickness. The cell was first filled with Protein A (Sigma P6031-1MG) solution (20 µl, 50 µg/ml) in BRB12 buffer (12 mM PIPES (pH 6.8), 2 mM MgCl$_2$, 1 mM EGTA) for 2 min. After washing unbound protein with 40 µl of BRB12 buffer, 20 µl of 20 µg/ml anti-α-tubulin antibody (Sigma T6199) solution in BRB12 buffer was infused for 5 min. Then unbound antibody was washed out with 40 µl of BRB12 buffer containing 20 µM taxol. Then, 40 µl of microtubule suspension in BRB12 buffer containing 20 µM taxol was infused and incubated for 2 min. Then, 40 µl of BRB12 buffer containing 20 µM taxol and 1 mg/ml casein was infused and incubated for 2 min. After washing with 40 µl of BRB12 buffer containing 20 µM taxol, the chamber was finally filled with observation buffer containing an ATP regenerating system (10 µg/ml creatin kinase, 2 mM creatin phosphate), an oxygen scavenging system (4.5 mg/ml glucose, 50 unit/ml glucose-oxidase, 50 unit/ml catalase, 0.5% 2-mercaptoethanol), 20 µM taxol, 200 nM Alexa Fluor 647 ATP (A22362; Invitrogen), and 100 pM kinesin in BRB12 buffer. All procedures were carried out at room temperature.

### Single-molecule FRET observation

Single-molecule observation of fluorescently labeled kinesin was done under a prism-type total internal reflection fluorescence microscopy (*Mori et al., 2007*). We used ATTO 488 (WT monomer) and Cy3 (others) as a donor fluorophore. ATTO 488 was excited with 488 nm diode laser (Cyan-75-TA-FE; Spectra-Physics Inc) and Cy3 was excited with 515 nm diode laser (EXLSR-515-50-CDTF-E, Spectra-Physics Inc). Fluorescent signals were collected through objective lens and a specific filter for cutting of the excitation light and projected on a cooled EMCCD camera. For smFRET observations, fluorescent signal was separated using dichroic mirror and donor and acceptor fluorescence were simultaneously observed. Position of microtubule was identified based on the fluorescence from the labeled kinesin. When more than five fluorescent spots of donor fluorophore lied on a straight line, we assumed that these spots were on the microtubule. FRET data were collected at 10- and 200-ms exposure times for wild-type monomer and for E236A monomer and E236A–WT heterodimer, respectively, at 22°C. We identified ATP binding to the kinesin head when the FRET efficiency exceeded a threshold of 0.5.

### Gold labeling of kinesin

Purified kinesin was biotinylated, and biotinylated kinesin was bound to microtubules in the presence of AMP-PNP as described previously (*Isojima et al., 2016*). Microtubule-bound kinesin was then collected by centrifugation (230,000 × *g* for 10 min) and was immersed in streptavidin solution containing 12 mM PIPES (pH 6.8), 2 mM MgCl$_2$, 1 mM EGTA, 20 µM taxol, and 2.5 mg/ml streptavidin (191-12851; Wako Pure Chemical Industries) and incubated 20 min at room temperature. Microtubule-bound streptavidin-modified kinesin was collected by centrifugation (230,000 × *g* for 10 min) and released from microtubules as described previously. Biotin-coated gold nanoparticles were prepared as previously described (*Isojima et al., 2016*), except that streptavidin was not added to the gold. Just before observation, streptavidin-modified kinesin and biotin-coated gold nanoparticles were mixed at a 1:1 molecule/particle ratio.

### Total internal reflection dark-field microscopy

A flow cell was constructed, and microtubules were attached on the glass surface by using protein A and anti-α-tubulin antibody as described previously (*Isojima et al., 2016*). The cell was then infused

with 40 µl of BRB12 buffer (12 mM PIPES (pH 6.8), 2 mM EGTA, and 1 mM MgCl$_2$) containing 20 µM taxol and 1 mg/ml casein for 2 min. The cell was infused with BRB12 buffer containing 3 nM gold-labeled kinesin for 5 min, and unbound kinesins and gold nanoparticles were removed by washing with 60 µl of the BRB12 buffer containing 20 µM taxol. Finally, the flow cell was infused with 40 µl of the observation buffer (BRB12 buffer containing 70 mM β-mercaptoethanol, 10 µg/ml creatine kinase, 2 mM creatine phosphate, 20 mM KCl, and 1–1000 µM ATP), sealed with nail polish and used for observation. All procedures were carried out at room temperature. The gold-labeled kinesin was observed using a total internal reflection dark-field microscope as described previously (*Isojima et al., 2016*) and the scattering images were recorded with a high-speed CMOS camera (FASTCAM Mini AX100; Photron) at a frame rate of 20,000 fps (50 µs temporal resolution). Observations were carried out at 24–26°C.

## Data analysis for the dark-field observations

The coordinates of gold nanoparticles were determined as described previously (*Isojima et al., 2016*). The orientations of the microtubule 'on axis' (orientation parallel to the microtubule long axis) and 'off axis' (orientation perpendicular to the microtubule long axis) were then determined from the aligned gold-labeled kinesin along the microtubule and adjusted to $X$ and $Y$ coordinates as described previously. The 'bound' and 'unbound' states of gold-labeled kinesin heads were determined as follows: two-dimensional standard deviation (s.d.) of the trajectory for each time frame $t$ was calculated as $[t - 20, t + 20]$ and the median $\mu$ and s.d. $\sigma$ of the s.d. of the trajectory was determined by fitting the histogram of the s.d. with a Gaussian function. Transitions from the bound to unbound state were detected when the s.d. was higher than $\mu + 4\sigma$ in 10 consecutive frames and transition from the unbound to bound state were detected when the s.d. was lower than $\mu + 4\sigma$ in 10 consecutive frames. The accurate frame of the transition was determined by fitting the on- and off-axis trajectory near the transition with two state step function. In the case of G0 construct, if two unbound states were closer than 50 ms and on-axis average of the bound state between these unbound states was separated from the other bound states, these unbound and bound states were determined as unbound states 1–3 (*Figure 4—figure supplement 2*). The rate constants of each state were determined by fitting the histogram of the dwell time with single exponential functions.

## Acknowledgements

We thank M Nakajima and Y Sakai for support with cloning. MT was supported by MEXT KAKENHI Grant Number 17H03661 and by JSPS KAKENHI Grant Numbers JP20H05542 and JP22H04850.

## Additional information

### Funding

| Funder | Grant reference number | Author |
| --- | --- | --- |
| Ministry of Education, Culture, Sports, Science and Technology | 17H03661 | Michio Tomishige |
| Japan Society for the Promotion of Science | JP20H05542 | Michio Tomishige |
| Japan Society for the Promotion of Science | JP22H04850 | Michio Tomishige |

The funders had no role in study design, data collection, and interpretation, or the decision to submit the work for publication.

### Author contributions

Yamato Niitani, Conceptualization, Software, Formal analysis, Investigation, Visualization, Methodology; Kohei Matsuzaki, Software, Formal analysis, Investigation, Visualization, Methodology; Erik Jonsson, Formal analysis, Investigation, Visualization, Methodology; Ronald D Vale, Supervision,

Funding acquisition, Methodology, Writing - review and editing; Michio Tomishige, Conceptualization, Supervision, Funding acquisition, Methodology, Writing - original draft, Project administration, Writing - review and editing

### Author ORCIDs
Ronald D Vale ⓘ https://orcid.org/0000-0003-3460-2758
Michio Tomishige ⓘ https://orcid.org/0009-0006-9182-4196

Reviewer #1 (Public review): https://doi.org/10.7554/eLife.106228.2.sa1
Reviewer #2 (Public review): https://doi.org/10.7554/eLife.106228.2.sa2
Reviewer #3 (Public review): https://doi.org/10.7554/eLife.106228.2.sa3
Author response https://doi.org/10.7554/eLife.106228.2.sa4

---

## Additional files

### Supplementary files
MDAR checklist

### Data availability
All data generated or analyzed during this study are included in the manuscript and supporting file; source data files have been provided for Figure 1—figure supplement 1 and 2, Figure 2, Figure 2—figure supplement 1–4, Figure 3—figure supplement 1, 2, and 4, Figure 4, Figure 4—figure supplement 1–3, Figure 5, Figure 5—figure supplement 1–3, Figure 6, Figure 6—figure supplement 1–5.

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
