## [Editor Report · eLife Assessment]

This study provides **compelling** evidence that kinesin's stepping mechanism is governed by strain-induced conformational changes in its nucleotide-binding pockets. Using pre-steady state kinetics and single-molecule assays, the authors demonstrate that the neck linker's conformation differentially modulates nucleotide affinity and detachment rates, establishing an asynchronous chemo-mechanical cycle that prevents simultaneous detachment. Supported by cryo-EM structural data, the work presents an **important** advance in our understanding of kinesin's hand-over-hand movement.

[Editors' note: this paper was reviewed by Review Commons.]

---

## [Referee Report · Reviewer #1 (Public review)]

Summary:

This manuscript investigates the role of the neck linker in coordinating the stepping cycles of the two heads of a kinesin-1 motor. Previous studies in the field showed that kinesin walks by alternating stepping of its heads, referred to as hand-over-hand. In this stepping mechanism, the front head of a kinesin dimer must remain bound until the rear head dissociates from the microtubule, moves forward, and rebinds to the tubulin on the plus-end side of the front head. There is a large body of work done to address this question. These studies all point to the central role of the 14 amino acid extension, a neck-linker, which connects the two heads to a common stalk, in coordination of kinesin motility. In a two-head-bound state, the motor domains (heads) are oriented parallel to the microtubule, but the neck linkers are orienting toward each other, thereby, breaking the symmetry in a homodimeric motor. In addition, the neck linkers are quite short, almost stretching to their near contour length to accommodate the microtubule binding of both heads. Previous studies pointed out that either the opposing orientation or the intramolecular tension of the neck linkers coordinate the stepping cycle.

However, we still do not know which step(s) in the chemo-mechanical cycle is controlled by the neck-linker to keep the two heads out of phase. The front head gating model postulates that ATP binding to the front head is gated until the rear head detaches from the microtubule. The rear head gating model proposes that the neck linker accelerates the detachment of the rear head from the microtubule. In this study, the authors use pre-steady state kinetics and smFRET to address this question. They measured ATP binding and microtubule detachment kinetics of kinesin's catalytic domain with neck linker constraints (1) imposed by disulfide crosslinking of the neck linker in monomeric kinesin in backward (rear head-like) and forward (front head-like) orientations, and (2) using the E236A-WT heterodimer to create a two-head microtubule-bound state with the mutant and WT heads occupying the rear and front positions respectively. They found that neck-linker conformation of the rear head reduces the ATP dissociation rate but has little effect on microtubule affinity. In comparison, the neck-linker conformation of the front head does not change ATP binding to the front head, but it reduces ATP-induced detachment of the front head, suggesting that a step after ATP binding (i.e. ATP hydrolysis or Pi release) is gated in the front head.

Significance:

I believe that this work will make an important contribution to the large body of literature focused on the mechanism of kinesin, which serves as an excellent model system to understand the kinetics and mechanics of a molecular motor. The mechanism proposed by the authors modifies the front-head gating model and is in agreement with recent structural work done on a kinesin dimer bound to a microtubule. Overall, the work is well performed, and the conclusions are well supported by the experimental data.

---

## [Referee Report · Reviewer #2 (Public review)]

Summary:

In this study, the authors investigate the molecular mechanism behind kinesin-1's coordinated movement along microtubules, with a focus on how ATP binding, hydrolysis, and microtubule attachment/detachment are regulated in the leading and trailing heads. Using pre-steady state kinetics and single-molecule assays, they show that the neck linker's conformation modulates nucleotide affinity and detachment rates in each head differently, establishing an asynchronous chemo-mechanical cycle that prevents simultaneous detachment. Supported by cryo-EM structural data, their findings suggest that strain-induced conformational changes in the nucleotide-binding pockets are crucial for kinesin's hand-over-hand movement, presenting a detailed kinetic model of its stepping mechanism. The manuscript is well-crafted, technically rigorous, and should be of significant interest to cell biology and cytoskeletal motor researchers.

Significance:

All conclusions are well-supported by the provided data. The findings address a critical gap in our understanding of how kinesin's two motor domains coordinate their movements, offering insights into the molecular basis of its stepping mechanism. This work should be of significant interest to the cytoskeletal research community.

Comments on latest version:

The authors have satisfactorily addressed my comments, although I recommend the addition of the following reference:

Lu Rao, Jan O. Wirth, Jessica Matthias, and Arne Gennerich. 2025. A Two-Heads-Bound State Drives KIF1A Superprocessivity. bioRxiv 2025.01.14.632505

This paper provides conclusive evidence that kinesin-1 predominantly adopts a one-head-bound state at limiting ATP concentrations and remains in this state for a significant portion of its enzymatic cycle even at saturating ATP. This limits its processivity compared to KIF1A, which predominantly adopts a two-heads-bound state under saturating ATP conditions. These findings directly support the authors' conclusion that trailing head dissociation is favored over leading head detachment.

---

## [Referee Report · Reviewer #3 (Public review)]

Kinesin-1 is a dimeric motor protein that transports cargo along microtubules. Its movement relies on the ability of its two catalytic motor domains (heads) to couple microtubule interactions with directional conformational changes and ATP turnover in a coordinated, alternating manner. The kinetics of these processes in each head are tightly regulated (gated) to ensure that at least one motor domain remains bound to the microtubule at all times, preventing detachment.

Niitani et al. investigated the gating mechanism by focusing on the role of the neck linker, a flexible region extending from the motor domain's C-terminus that undergoes conformational changes during stepping. They examined how the neck linker differentially regulates the microtubule affinity and ATP turnover of the front and rear heads. To do this, they designed cross-linkable monomeric motor domains mimicking the conformations of the front and rear heads and employed a combination of pre-steady-state and single-molecule analyses to measure ATP-binding and microtubule-detachment kinetics. Additionally, they studied a kinesin heterodimer with a locked rear head conformation to distinguish the kinetic properties of the front and rear heads within an active dimer.

ATP binding rates were measured using stopped-flow experiments with mant-ATP and nucleotide-free kinesin-microtubule complexes. The results showed that crosslinking the neck linker in the forward-pointing conformation (mimicking the rear head) reduced the ATP dissociation rate, while crosslinking it in the rear-pointing conformation (mimicking the front head) had no significant effect on ATP binding kinetics. ATP dissociation from the rear head was further examined using a kinesin mutant (E236A) that stabilizes the ATP-bound state by significantly slowing ATP hydrolysis.

To assess how neck-linker orientation affects microtubule attachment, the authors monitored turbidity changes after rapidly mixing nucleotide-free, crosslinked kinesin-microtubule complexes with ATP in a stopped-flow apparatus. Their findings demonstrated that the forward-oriented neck linker in the rear head promotes microtubule detachment, whereas the backward-oriented neck linker in the front head reduces detachment rates.

These results indicate that neck-linker conformation governs gating of microtubule affinity and nucleotide binding. Moreover, they show that even partial docking of the neck linker onto the head is sufficient to partially open the gating mechanism. To further investigate the role of neck linker tension, the authors created kinesin dimers with neck linker insertions of varying lengths. Microtubule detachment kinetics and ATPase activity assays revealed that ATP turnover in the rear head is significantly affected by the degree of forward tension applied to its neck linker.

Overall, Niitani et al. build upon previous kinesin gating models by introducing a neck-linker tension-based ATP binding affinity mechanism. Their findings provide a mechanistic basis for recent cryo-EM observations for kinesin-1 and kinesin-3 (KIF14) and distinguish the specific roles of neck linker tension in the front and rear heads in regulating ATP binding, hydrolysis, and microtubule detachment. This study is biochemically rigorous and makes an important contribution, though direct structural validation (e.g., cryo-EM snapshots of crosslinked or mutant kinesins bound to microtubules) would further strengthen their conclusions and clarify the asymmetry in ATP affinity between the front and rear heads.

---

## [Author Response]

We thank the reviewers for the detailed evaluations and thoughtful comments, which have improved the clarity and readability of this manuscript. We have responded to all reviewer comments and incorporated their suggested changes into the text and figures. We have also included new experimental results suggested by reviewer 2, which further strengthen our main conclusion.

Point-by-point description of the revisions

**Reviewer #1:**
(1) Introduction, page 3: The statement "Single dimeric kinesin moves processively along microtubules in a hand-over-hand manner by alternately moving the two heads in an 8-nm step toward the plus-end of the microtubule" is inaccurate. The kinesin heads take ~16 nm steps, while the center of mass advances in ~8 nm increments. Please adjust the wording accordingly.(2) Introduction, page 5: In the sentence "These results are consistent with the closed and open conformations of the nucleotide-binding pocket in the rear and front heads of microtubule-bound kinesin dimers observed in cryo-electron microscopy (cryo-EM) studies," I recommend changing the order to align with the previous sentence. The correct order would be "These results are consistent with the open and closed conformations of the nucleotide-binding pocket in the front and rear heads."

We thank the reviewer for pointing out our misunderstandings. We have corrected these sentences accordingly (lines 45-47 and lines 111-112).

**Reviewer #2:**
MAJOR CONCERNSLimitations of this study: The authors need to discuss the limitations of their work. (1) They used a cys-lite kinesins mutant and introduced new surface-exposed cysteines. These mutants have lower kcat values than WT. (2) They used fluorescently labeled ATP molecules, which are hydrolyzed 10 times slower than unlabeled nucleotides. (3) They still observe crosslinking under reducing conditions and partial (but almost complete) crosslinking under oxidized conditions. (4) They assumed that cysteine crosslinked orientation mimics the orientation of the neck-linker in the front and rear conditions. The authors clearly pointed to these issues in the Results section. While these assumptions are also supported by several control experiments, the authors need to acknowledge some of these limitations in the Discussion as well.

We have now reiterated some of the key caveats in the Discussion, and newly described in the Results section those points not mentioned in the original manuscript that do not affect the conclusion. We also added a summary of the limitations and caveats into the first paragraph of the Discussion section (lines 425-431).

(1) We added a sentence in the Results section to describe that the ATP-binding kinetics of the Cys-light mutant remained consistent with previous studies as follows: “First, we demonstrated that *k+1* and *k-1* of the wild-type head without Cys-modification were unchanged after oxidization (Table 1) and were comparable to those previously reported (Cross, 2004)” (lines 163-166). The reduced kcat values of cysteine pair-added mutants before crosslinking were primarily due to reduced microtubule association rate (data not included in this manuscript). We have added a sentence in the Results section describing the kcat results as follows: “The reduced ATPase activity primarily results from a decreased microtubule association rate (data to be presented elsewhere) with little change in ATP binding or microtubule dissociation rates (Table 1).” (lines 144-146).

(2) Fluorescently-labeled ATP was used to determine the ATP off-rates of the E236A mutant monomer and E236A rear head of the E236A/WT heterodimer. Two caveats in these measurements could lead to underestimating the ATP off-rate: (1) The off rate of Alexa-ATP from the head may be reduced compared to unmodified ATP, as Alexa-ATP driven motility showed a 10-fold reduce velocity. (2) The ATP off-rate of the E236A mutant may differ from that of the rear head in the wild-type dimer, since the E236A mutant likely stabilizes the neck linker-docked state more strongly than in the rear head of the wild-type dimer. These points are crucial for evaluating the results of ATP off-rate and the affinity for ATP, so we have added sentences in the Discussion section as follows: “We note, however, that this *Kd* of ATP may somewhat underestimate the true value in wild-type kinesin for two reasons: first, the E236A mutation likely stabilizes the neck linker-docked, closed state more than in the rear head of the wild-type dimer (Rice et al., 1999), and second, the Alexa-ATP used to measure the ATP off-rate of E236A head showed ~10-fold smaller velocity compared to unmodified ATP, partly due to a slower ATP off-rate (Figure 2-figure supplement 3).” (lines 449-454).

(3) Under reducing condition, the rear head crosslink contained 30% crosslinked species, while under oxidized condition, the front head crosslink contained 11% un-crosslinked species (Figure 1-figure supplement 1). These heterogeneities likely affect the rate constants of *K-1* for rear head crosslink and *K2* for front head crosslink, as crosslinked and un-crosslinked species showed significantly different rate constants. However, we did not use the rear head crosslink result to determine *K-1*, since ATP hydrolysis likely occurred before reversible ATP dissociation. Instead, we used E236A monomer to estimate the *K-1* of the rear head. In addition, the result for *K2* of the front head crosslink was further validated using the E236A/WT heterodimer, which will be described in the next section.

(4) This is an important point, and therefore, we conducted experiments using the E236A/WT heterodimer (including new experimental results of ATP binding kinetics of the front head) and obtained consistent results. To address this point, we have revised the following sentences in the Discussion: “In the front head, backward orientation of the neck linker has little effect on ATP binding and dissociation rates, both when measured for a monomer crosslink (Figure 2A, B) and for the front head of a E236A-WT heterodimer (Figure 4B, C, F).” (lines 432-433); “However, we found that the ATP-induced detachment rates from microtubule (*K2*) were similarly reduced for both the front head crosslink (7.0 s^-1^; Figure 3A) and the front WT head of the E236A/WT heterodimer (6.3 s^-1^; Figures 6D), suggesting that a step subsequent to ATP binding is gated in the front head.” (lines 437-441).

Line 238, the authors wrote that "forward constraint on the neck linker in the rear head does not significantly accelerate the detachment from the microtubule." Can the authors comment on why the read-head-like construct has a low affinity for microtubules even in the absence of ATP (Line 220)? I believe that the low affinity of the head in this conformation is more striking (and potentially more important) than the changes they observe in detachment rates. The authors should also consider that they might not be able to reliably measure the changes in the dissociation rate in single molecule assays of this construct (especially if the release rate of the rear head in the oxidized condition increases a lot higher than that of WT). The kymographs show infrequent and brief events, which raises doubts about how reliably they can measure the release rates under those imaging conditions. Higher motor concentrations and faster imaging rates may address this concern.

The low microtubule affinity of the rear-head-like crosslink stems from an extremely slow ADP release rate upon microtubule binding, not from a fast microtubule-detachment rate. Using stopped-flow measurements of microtubule-binding kinetics (microtubule-stimulated mant-ADP release and microtubule association rates), we found that the rear-head-crosslink resulted in a 2,000-fold decrease in the microtubule-stimulated ADP-release rate. This finding also explains the reduced ATPase of the rear-head-crosslink (Figure 1E). Since this low microtubule-affinity state occurs in the ADP-bound state rather than the ATP-bound state, we hypothesized that the neck-linker docked ADP-bound state cannot effectively bind to microtubules, requiring neck-linker undocking for microtubule binding (Mattson-Hoss et al., Proc. Natl. Acad. Sci., 111, 7000-7005 (2014)). While we acknowledge that understanding slow microtubule binding in the neck linker docked state is important for elucidating the mechanism and regulation of microtubule-binding of the head, this paper focuses specifically on the mechanism and regulation of “microtubule-detachment”. We plan to present these microtubule-binding kinetics data in a separate manuscript currently in preparation.

To explain the low microtubule affinity of the rear-head-crosslink, we added this explanation to the text; “because this constraint on the neck linker dramatically reduces the microtubule-activated ADP release rate (data to be presented elsewhere), creating a weak microtubule binding state” (lines 226-228).

Although the rear head crosslinking construct under oxidative condition showed fewer fluorescent spots per kymographs (images) due to its low microtubule binding rate, we collected more than one hundred spots by recording additional microscope movies (N=140; Figure 3-figure supplement 2B), ensuring sufficient data for statistical analysis.

Figure 2: How do the rates shown in Figure 2A-B compare to the previous kinetics studies in the field? The authors compare the dissociation rate of WT measured in rapid mixing experiments to that of E236A in smFRET assays. It is not clear whether these comparisons can be made reliably using different assays. Can the authors perform rapid mixing of E236A or try to determine the rate for the WT from smFRET trajectories?

The results of ATP on/off rates are comparable to the previous stopped flow measurements of ATP binding to monomeric kinesin-1 on microtubule, which are 2-5 µM^-1^s^-1^ and ~150 s^-1^, respectively (summarized in the review by Cross (2004)). We added a sentence as follows: “First, we demonstrated that *K+1* and *K-1* of the wild-type head without Cys-modification were unchanged after oxidization (Table 1) and were comparable to those previously reported (Cross, 2004).” (lines 163-166).

As the reviewer pointed out, the rapid mixing and smFRET data cannot be directly compared due to the differences in temporal resolution and fluorescent probe used. In Figure 2E (2F in the revised version), we measured ATP dissociation rate for both WT and E236A using smFRET. Due to the lower temporal resolution, we could not accurately determine ATP binding rate using smFRET. Therefore, to compare the ATP binding rate between WT and E236A heads, we now have added stopped-flow measurements of mant-ATP binding to the E236A monomer, as shown in Fig. 2C and Figure 2-supplement 2, and described in the text (lines 182-185).

Line 396: One of the most significant conclusions of this work is that the backward orientation of the neck linker has little effect on ATP binding to the front head. This is only supported by the results shown in Fig. 2A-B. Can the authors perform/analyze smFRET assays on the E236A/WT heterodimer to directly show whether the ATP binding rate to the WT head is affected or not affected by the orientation of the neck linker of the WT head?

We agree with the reviewer that our finding about ATP binding to the front head is potentially significant in the kinesin field, as it has been widely believed that ATP-binding is suppressed in the front head. In our original manuscript, this conclusion was supported only by the measurement of ATP on-rate of the front-head-crosslink, which may differ from the front head of a dimer in which the backward orientation of the neck linker is maintained by the backward strain. Although the reviewer suggested performing smFRET experiments using E236A/WT heterodimer, smFRET have relatively low temporal resolution (50-100 fps) and cannot accurately measure the frequency of ATP binding, so we used this technique only to determine ATP off rates. In this revised manuscript, we now have added stopped-flow experiments to separately measure the ATP binding to the front and rear heads of the E236A/WT heterodimer. By labeling the rear E236A head with a fluorophore to quench the mant-ATP signal bound to the rear head, we successfully measured mant-ATP binding rate to the front head. We found that the ATP-binding rate to the front head was comparable to that of an unconstrained monomer head, providing direct evidence for our conclusion. The revised version includes Fig. 4 A-C (with Figure 4-supplement 2; Figs. 4 and 5 are swapped in order) showing the kinetics of ATP binding to the front and rear heads of the E236A/WT heterodimer, with corresponding text in the result section (lines 315-324).

MINOR CONCERNSLines 31 and 32: I recommend replacing "ATP affinity" with "ATP binding rate" or "the dissociation of ATP" to be more specific. This is because they do not directly measure the affinity (Kd), but instead measure the on or off rates.Line 41: Replace "cellar" with "cellular".Line 83: The authors should cite Andreasson et al. here.

We have corrected these sentences accordingly (lines 31, 40, 85).

Lines 83-86: It seems this sentence belongs to the next paragraph. It also needs a citation(s).

This statement lacks experimental evidence and may confuse readers, so we have removed it for clarity.

Line 151: It would be helpful to add a conclusion sentence at the end of this paragraph to explain what these results mean to the reader.

A conclusion sentence of this paragraph has been added: “These results demonstrate that neck linker constraints in both forward and rearward orientations inhibit specific steps in the mechanochemical cycle of the head (lines 151-153)”.

Lines 175-180: I recommend combining and shortening these sentences, as follows, to avoid confusing the reader: "To detect the ATP dissociation event of the rear head, we employed a mutant kinesin with a point mutation of E236A in the switch II loop, which almost abolishes ATPase hydrolysis and traps in the microtubule-bound, neck-linker docked state,"

We have corrected these sentences accordingly (line 179-181).

Line 314: "which was rarely observed ...". This is out of place and confusing as is. I recommend moving this sentence after the sentence that ends in Line 295.

This sentence explains how the dark-field microscopy data was analyzed to determine whether the labeled head was in the leading or trailing position before detaching from the microtubule, but the explanation needs clarification. We removed the phrase “which was rarely observed for E236A-WT heterodimer” and simplified this sentence as follows: “Moreover, these observations allow us to distinguish whether the gold-labeled WT head was in the leading or trailing position just before microtubule detachment; the backward displacement of the detached head indicates that the labeled WT head occupied the leading position prior to detachment (Figure 5-figure supplement 1).” (lines 347-351).

Line 300: Can the authors comment on why E236A/WT has a substantially lower ATPase rate than WT homodimer? Is it possible to determine which step in the catalytic cycle is inhibited?

We demonstrated that the *k2* (microtubule-detachment rate) of the front head matched the ATP turnover rate of the E236A/WT heterodimer (Figure 6 B and E), suggesting that the inhibited step occurs after ATP binding in the front head. In contrast, the rear E236A head showed virtually no ATP hydrolysis activity, since in high-speed dark field microscopy, we observed forward step caused by rear E236A head detachment from microtubule only rarely, approximately once every few seconds (Figure 5-figure supplement 1). We added a sentence in the text as follows: “As described later, the reduced ATPase rate results from suppressed microtubule detachment of the front WT head, while the rear E236A head is virtually unable to detach from microtubules” (lines 311-313).

Line 323: Is the unbound dwell time unchanged?

The unbound dwell time exhibited a weak ATP-dependence, which we described only in Figure 5-supplement 2 (Figure 4-supplement 2 in the old version). We observed three distinct phases in the unbound dwell time based on mobility differences, with ATP dependence appearing only in the third phase. This finding suggests that ATP binding to the microtubule-bound E236A head is sometimes necessary for the detached WT head to rebind to the forward-tubulin binding site, indicating that the microtubule-bound E236A head occasionally releases ATP during the one-head-bound state (without the forward neck linker strain). To describe the ATP-dependence of the unbound dwell time, we added a sentence in the main text as follows: “In contrast, the dwell time of the unbound state of the gold-labeled WT head showed weak ATP dependence (Figure 5-figure supplement 2), indicating that the rear E236A head occasionally releases ATP when the front head detaches from the microtubule and the neck linker of E236A head becomes unconstrainted. This finding further supports the idea that forward neck linker strain plays a crucial role in reducing the reversible ATP release rate.” (lines 372-377).

Line 331: I recommend replacing "ATP-induced detachment" with "nucleotide-induced detachment" for clarity.

We have revised the phrase accordingly (line 371).

Line 344: I recommend replacing "affinity" with "forward strain prevents the release of the nucleotide" or similar to avoid confusion. Forward strain reduces the off-rate of the bound nucleotide, rather than allowing ATP to bind more efficiently to the rear head.

We agree to the reviewer’s comment and have corrected this sentence accordingly (line 338).

Lines 376-385: G7-12 constructs are introduced in Figure 6, but the results in this paragraph are shown in Figure 5. They should be moved to Figure 6 to avoid confusion.

To improve the readability, we have reorganized Figures 4-6, such that all the figure panels related to the neck linker extended mutants are shown in Figure 6; Figure 5D has been moved to Figure 6F.

Line 421: delete "not" before "does not".

We have corrected this typo.

Lines 433-441: Unless I am mistaken, more recent work in the kinesin field showed that backward trajectories of kinesin 1 reported by Carter and Cross are due to slips from the microtubule rather than backward processive runs of the motor.

The slip motion demonstrated by Sudhakar et al. (2021) differs from the backstep motion reported by Carter and Cross (and many other laboratories). Slip motion occurs after kinesin detaches from the microtubule and continues until the bead returns to the trap center. In contrast, backstep motion occurs during processive movement when the trap force either exceeds or approaches the stall force. The kinetics of these motions also differ significantly: slip steps occur with a dwell time of 71 µs and are independent of ATP concentration, while backsteps take ~0.3 s (at 1 mM ATP) and depend on ATP concentration. These differences indicate that slip motion is phenomenologically distinct from backsteps occurring under supra-stall or near-stall force.

Line 474: Replace "suppresses" with "suppressed".

We have corrected this typo.

Figure 4E: I would plot these results with increasing ATP concentration on the x-axis.

We formatted Figure 4E to match Figure 4b from Isojima et al. (Nature Chem. Biol. 2015), to emphasize the difference in ATP dependence of the front and rear head.

Figure 4B: The authors should explain how they distinguish between bound and unbound states in the main text or figure legends. For example, it is not clear how the authors score when the motor rebinds to the microtubule in the first unbinding event shown in Figure 4B (displacement plot).

The method was described in the Materials and Methods section, but we have now described how to distinguish between bound and unbound states in the main text as follows: “Unlike the unbound trailing head of wild-type dimer that showed continuous mobility (Isojima et al., 2016), the unbound WT head of E236A-WT heterodimer exhibited a low-fluctuation state in the middle (Figure 5B, s.d. trace). This low-fluctuation unbound state was distinguishable from the typical microtubule-bound state, having a shorter dwell time of ~5 ms compared to the bound state and positioning backward, closer to the E236A head, relative to the bound state (Figure 5-figure supplement 2).” (lines 351-356).

**Reviewer #3:**
Minor Issues:- Line 22, Abstract - The phrase "move in a hand-over-hand manner" could be clearer if phrased as "move in a hand-over-hand fashion" to improve readability.

We changed the word “manner” to “process” (line 23).

- Abstract - Neck linker conformation in the leading head: The sentence "We demonstrate that the neck linker conformation in the leading kinesin head increases microtubule affinity without altering ATP affinity" would benefit from defining this conformation as "backward" for clarity.- Abstract - Neck linker conformation in the trailing head: The sentence "The neck linker conformation in the trailing kinesin head increases ATP affinity by several thousand-fold compared to the leading head, with minimal impact on microtubule affinity" should also clarify that this conformation is "forward."

We have corrected these sentences accordingly (line 30, 32).

- Abstract - Conformation-specific effects: The authors mention conformation-specific effects in the neck linker structure but do not define the neck linker's conformation or the motor domain's (MD) conformation. Clarifying these conformational changes would improve the explanation of how they promote ATP hydrolysis and dissociation of the trailing head before the leading head detaches from the microtubule, thereby providing a kinetic basis for kinesin's coordinated walking mechanism.

We have revised the last sentence of the abstract accordingly by specifying the neck linker’s conformation as follows: “In combination, these conformation-specific effects of the neck linker favor ATP hydrolysis and dissociation of the rear head prior to microtubule detachment of the front head, thereby providing a kinetic explanation for the coordinated walking mechanism of dimeric kinesin.” (lines 34-37).

- Line 306 - Use of ATP in the E236A-WT heterodimer: In discussing the "ATP-induced detachment rate of the WT head in the E236A-WT heterodimer," the authors should consider justifying their choice of ATP over ADP for inducing microtubule (MT) dissociation. Since ATP typically promotes tighter MT binding and ATP turnover is reduced in forward-positioned WT heads, it may be unclear to some readers why ATP was chosen.

We measured the ATP-induced detachment rate *k2* of the front head of the E236A-WT heterodimer to validate our findings from the front-head-crosslinked monomer experiments, which demonstrated reduced *k2* after oxidation. To clarify this point, we have now included ATP binding kinetics measurements for both front and rear heads of the E236A-WT heterodimer, as suggested by reviewer 2. These additional data demonstrate consistency between the results from the crosslinked monomer and E236A-WT heterodimer experiments.

- Discussion - Backward-oriented neck linker in the front head: The discussion mentions that the backward-oriented neck linker in the front head reduces its ATP-induced detachment rate, suggesting that a step after ATP binding (e.g., isomerization, ATP hydrolysis, or phosphate release) is gated in the front head. However, the authors do not clarify that the backward neck linker orientation would imply the nucleotide pocket should be open or at least not fully closed, thus inhibiting ATP turnover. This is important because, as demonstrated in other studies, full closure of the nucleotide pocket is linked to neck linker docking. This point should be addressed earlier in the discussion.

We have addressed this point by revising this sentence as follows: “These results are consistent with an inability of the front head to fully close its nucleotide pocket to promote ATP hydrolysis and Pi release (Benoit et al., 2023), as will be discussed later.” (lines 441-443)